behaviour, ecology, evolution

sensory ecology, plant–animal interactions, colour vision, olfaction, frugivory, platyrrhine

**Authors for correspondence:**
Amanda D. Melin
e-mail: amanda.melin@ucalgary.ca
Carrie C. Veilleux
e-mail: cveill@midwestern.edu
Shoji Kawamura
e-mail: kawamura@edu.k.u-tokyo.ac.jp

†Indicates equal contributions.

# Anatomy and dietary specialization influence sensory behaviour among sympatric primates

Amanda D. Melin[1,2,3,4,†], Carrie C. Veilleux[5,6,†], Mareike C. Janiak[1,3,7], Chihiro Hiramatsu[8], Karem G. Sánchez-Solano[9], Ingrid K. Lundeen[6], Shasta E. Webb[1], Rachel E. Williamson[1], Megan A. Mah[1], Evin Murillo-Chacon[10], Colleen M. Schaffner[11], Laura Hernández-Salazar[9], Filippo Aureli[9,12] and Shoji Kawamura[13]

[1]Department of Anthropology and Archaeology, and [2]Department of Medical Genetics, University of Calgary, Calgary, AB, Canada
[3]Alberta Children's Hospital Research Institute, University of Calgary, Calgary, AB, Canada
[4]German Primate Research Center, Gottingen, Germany
[5]Department of Anatomy, Midwestern University, Glendale, AZ, USA
[6]Department of Anthropology, University of Texas, Austin, TX, USA
[7]School of Science, Engineering & Environment, University of Salford, Manchester, UK
[8]Department of Human Science, Faculty of Design, Kyushu University, Fukuoka 815-8540, Japan
[9]Instituto de Neuroetología, Universidad Veracruzana, Xalapa, Veracruz, México
[10]Área de Conservación Guanacaste, Guanacaste, Costa Rica
[11]Psychology Department, Adams State University, Alamosa, CO, USA
[12]Research Centre in Evolutionary Anthropology and Palaeoecology, Liverpool John Moores University, Liverpool, UK
[13]Department of Integrative Biosciences, University of Tokyo, Kashiwa, Japan

ADM, 0000-0002-0612-2514; CH, 0000-0003-1191-4476; SEW, 0000-0002-9329-2553; REW, 0000-0001-5414-5799; LH-S, 0000-0001-7567-8068; SK, 0000-0003-0350-6050

Senses form the interface between animals and environments, and provide a window into the ecology of past and present species. However, research on sensory behaviours by wild frugivores is sparse. Here, we examine fruit assessment by three sympatric primates (*Alouatta palliata*, *Ateles geoffroyi* and *Cebus imitator*) to test the hypothesis that dietary and sensory specialization shape foraging behaviours. *Ateles* and *Cebus* groups are comprised of dichromats and trichromats, while all *Alouatta* are trichromats. We use anatomical proxies to examine smell, taste and manual touch, and opsin genotyping to assess colour vision. We find that the frugivorous spider monkeys (*Ateles geoffroyi*) sniff fruits most often, omnivorous capuchins (*Cebus imitator*), the species with the highest manual dexterity, use manual touch most often, and that main olfactory bulb volume is a better predictor of sniffing behaviour than nasal turbinate surface area. We also identify an interaction between colour vision phenotype and use of other senses. Controlling for species, dichromats sniff and bite fruits more often than trichromats, and trichromats use manual touch to evaluate cryptic fruits more often than dichromats. Our findings reveal new relationships among dietary specialization, anatomical variation and foraging behaviour, and promote understanding of sensory system evolution.

## 1. Introduction

Sensory systems play crucial roles in food investigation, both in finding food and evaluating food quality [1,2]. A large body of work has explored the role of colour vision in finding and assessing foods in terrestrial vertebrate taxa [3–5] but the role of non-visual senses during foraging has received

considerably less attention, with the notable exception of chiropterans [6], and a growing literature on primates and scatter-hoarding rodents [7–9]. Despite this, a growing body of literature suggests non-visual senses are critical to food investigation [10–12]. Many animals routinely sniff fruits [13–15] and variation in sniffing behaviour has been attributed to changes in fruit odour profile during ripening [8]. Fleshy fruits also typically become softer as they mature, offering a potential haptic cue of fruit quality [10,16].

Sensory systems are well documented to be responsive to selective pressures, and interspecific variation in anatomy has been used to make judgements about the relative importance of different senses. For example, high mechanoreceptor density and size in the trigeminal ganglia of ducks have been associated with tactile foraging [17], while the large olfactory bulbs of seabirds are thought to reflect emphasis on olfaction for foraging and navigation [11,18]. The relatively large main olfactory bulbs (MOBs) and higher densities of mechanoreceptors in the fingers of frugivorous primates have similarly been interpreted as evidence that olfaction and manual touch, respectively, are important for fruit foraging [19–21]. However, little comparative work has explored variation in sensory behaviour among wild, sympatric species and associations with their dietary specialization or anatomy. This is particularly true for non-visual senses, e.g. olfaction, taste and touch.

Primates occupy a wide variety of dietary niches, possess notable variation in sensory systems and are relatively easy to observe in wild foraging contexts [1,22,23]. Although sparse, current literature suggests that primates differ in sensory use [10,24–26]. For example, when presented with the same novel stimuli, spider monkeys (*Ateles geoffroyi*) were more likely to sniff stimuli, while squirrel monkeys (*Saimiri sciureus*) were more likely to touch them [25]. Notably, primates exhibit intraspecific and interspecific colour vision variation. Unlike most mammals, which are primarily dichromatic (colour vision based on two cone types, red-green 'colourblind'), many primate species possess trichromatic colour vision (based on three cone types), and the capacity to distinguish reds from greens (reviewed in [27]). While catarrhine primates (African and Asian monkeys and apes, including humans) exhibit routine trichromacy, most platyrrhines (monkeys in the Americas) exhibit polymorphic trichromacy [27,28]. In this system, allelic variation at the single X-linked M/L (*OPN1LW*) opsin gene results in trichromacy in heterozygous females but dichromacy in males and homozygous females, leading to multiple vision types in the same social group [27,28]. By contrast, howler monkeys (genus *Alouatta*) have independently evolved routine trichromacy through an opsin gene duplication, similar to—but independent from—the evolution of routine trichromacy in catarrhines [28]. Trichromacy has been hypothesized to confer advantages for detecting reddish ripe fruits against green leaves [29–31]. Differences in colour vision abilities can also influence the use of other senses, including olfaction [8,15], and trichromats have been recorded to have a higher acceptance index during foraging, i.e. to reject fewer of the fruits they handle [15].

Most studies of sensory behaviour to date have involved laboratory- or field-based experiments; comparative studies of sensory investigation in wild animals foraging under natural conditions are largely absent. This lack of behavioural data complicates research employing anatomical 'proxies'

for sensory reliance. For example, although larger MOBs, olfactory nasal epithelial surface areas, or ethmoid bones are suggestive of increased olfactory importance or sensitivity [19,21,32–35], it is unclear whether species with these features sniff foods more frequently. Similar questions can be raised for the other senses; for example, is the density of fungiform papillae on the tongue, which has been linked to taste sensitivity [36], associated with tasting behaviours, such as biting/rejecting or licking?

Here, we examine three species of wild sympatric primates to address two questions: (1) Do primate species vary in their fruit investigative behaviours reflecting differences in dietary specialization or sensory anatomy? We hypothesize that dietary specialization for frugivory shapes foraging behaviours as an adaptation to assess fruit quality, and investigate whether commonly used morphological proxies are associated with behavioural reliance on different senses. We predict more frugivorous species use sensory investigation (smell, touch and taste) more often during fruit selection. We also predict that species with (i) larger MOB volumes and olfactory turbinate surface areas sniff more frequently, (ii) higher densities of fungiform papillae on the tongue use gustation (as proxied by biting and rejecting) more frequently and (iii) higher thumb to index finger ratios (a proxy for higher manual dexterity) use manual touch more frequently. (2) Does variation in colour vision phenotype (dichromatic versus trichromatic vision) influence the use of other sensory behaviours and food selection accuracy? Because two of the three primate species exhibit intraspecific colour vision variation via polymorphic opsin genes, we can investigate the relationships between colour vision and non-visual senses, while controlling for species-level variation. We predict that dichromatic monkeys use non-visual senses more frequently than trichromats during food investigation, and that monkeys with trichromatic colour vision are more accurate in fruit selection, leading to higher acceptance indices relative to dichromatic individuals for conspicuous fruits that undergo a long wavelength (reddish) colour change during ripening.

## 2. Methods

### (a) Study species and behavioural data collection

We studied three sympatric species with divergent diets living in the seasonal tropical dry forest of Sector Santa Rosa (SSR), Área de Conservación Guanacaste (ACG) in northwestern Costa Rica (10°45′–11°00′ N, 85°30′–85°45′ W): Geoffroy's spider monkeys (*Ateles geoffroyi*), white-faced capuchins (*Cebus imitator*) and mantled howler monkeys (*Alouatta palliata*) (figure 1). Primates in Santa Rosa have been studied for decades and are well habituated to human observers [37]. To supplement the SSR *Alouatta palliata* data, for which we had the fewest records, we also studied a population of *Alouatta palliata* in Isla Agaltepec, Mexico [38,39]. Observation details are presented in the electronic supplementary material, Methods S1 and table S1.

We conducted short (1–10 min) continuous focal animal samples following a published protocol [22] with strict out-of-site rules, such that we only recorded behaviour when we had an unobstructed view of the focal monkey's hands and face. Individuals were sampled opportunistically, based on visibility, but we rotated among sex and age classes in an effort to sample evenly across these variables. We recorded fruit investigation sequences, including each manual touch, sniff and bite event,

**Figure 1.** Study species and principal diet: (a) black-handed spider monkey, *Ateles geoffroyi*, frugivore; (b) white-faced capuchin monkey, *Cebus imitator*, (frugivore–omnivore) and (c) mantled howler monkey, *Alouatta palliata*, (folivore–frugivore). Photo credits: Fernando Campos (a), Amanda Melin (b,c). (Online version in colour.)

and whether the investigated fruit was eaten or rejected. Sniffing was coded as present (yes/no) if fruits were brought close to or in contact with the nose (electronic supplementary material, video S1 and S2). 'Bite' as a sensory assessment was only recorded when the fruit was rejected, as mastication is a requirement of fruit consumption and we cannot tease apart feeding and assessment for consumed fruits. We classified a fruit as eaten if at least two bites were taken.

## (b) Dietary specialization

To quantify the relative amounts of frugivory for *Ateles geoffroyi*, *Cebus imitator* and *Alouatta palliata*, we conducted a literature review to determine the monthly range in dietary proportion composed by fruits for each species (electronic supplementary material, dataset S1). We searched for studies on Google Scholar using the terms 'Ateles geoffroyi', 'Cebus capucinus', 'Cebus imitator', 'Alouatta palliata', 'diet' and 'frugivory' without applying date restrictions. We included studies conducted throughout the geographical ranges of each species that used focal animal sampling to collect feeding data from study groups inhabiting relatively undisturbed forest, for a period of at least six months.

## (c) Sensory anatomy

We assembled data on the anatomy associated with olfaction, taste and manual touch (haptic sensation) from the literature or collected from skeletal material and images for the three species (or congeners when necessary) of platyrrhines we studied.

### (i) Olfaction

We used two anatomical proxies of olfactory sensitivity: MOB volume and nasal turbinate surface area. The MOB receives the projections of olfactory receptors (ORs) stimulated by food-associated odours and is involved in processing olfactory information [40,41]. We reviewed published literature to obtain measures of absolute MOB (aMOB) and relative MOB (rMOB) volume for the three species [42]. While the species differ in body mass and total brain volume, aMOB volume itself has been used as a metric of olfactory importance, as it represents a measure of the total number of olfactory neurons in a given individual/species [41]. We also included rMOB size, calculated as per cent of the total brain volume. Nasal turbinate surface area was measured as an anatomical correlate of the size of the nasal epithelium which contains the ORs [40]. We measured the posterior/superior row of turbinates that are covered in olfactory

epithelium *in vivo* (ethmoturbinate and nasoturbinate [43]) using computed tomography (CT) scans [44]. We downloaded CT scans (*n* = 8 *Alouatta palliata*, 15 *Ateles geoffroyi*, two *Cebus imitator* previously *C. capucinus*)) from the digital repository Morphosource.org (electronic supplementary material, dataset S2). Using Avizo 9.1® (http:// www.fei.com/software/avizo3d/), we digitally extracted olfactory turbinates from each specimen, visualized turbinates as three-dimensional surfaces and measured the surface area (electronic supplementary material, figure S1). We also collected data on skull size (calculated as the geometric mean of cranial length and width) for each specimen.

### (ii) Taste

Fungiform papillae in the tongue house taste receptors, and increased density of these papillae on the anterior tongue is positively associated with taste sensitivity [36]. We obtained published data [45] on the density of fungiform papillae on the anterior 0.5 cm of the tongue and body mass for 14 howler (*Alouatta palliata*), four spider (*Ateles geoffroyi*) and 12 capuchin (*Sapajus* [*Cebus*] *apella*) monkeys (data were not available for *Cebus imitator*).

### (iii) Manual touch

Data on anatomical proxies for the haptic sense are scarce. We use the ratio of thumb to index finger length, an established measure of manual dexterity [46], as a proxy for the importance of manual manipulation and sensation. We collected data on thumb and index finger lengths from digital pictures of hands belonging to the study species or congeneric species using ImageJ [47]. *Ateles geoffroyi* exhibits intraspecific variation in the presence of a vestigial external thumb (electronic supplementary material, figure S2). Individuals lacking external thumbs were assigned a thumb : index ratio of zero. All ImageJ measurements were performed twice each by two authors (A.D.M. and M.A.M.). We supplemented the data we collected with either published digit lengths [46] or published thumb : index ratios [48] for other *Ateles, Alouatta* and *Cebus* individuals (electronic supplementary material, dataset S3).

## (d) Colour vision type

The colour vision systems of our study species are well established [27]. *Cebus* and *Ateles* have polymorphic colour vision with both dichromats (males and homozygous females) and trichromats (heterozygous females) in their populations. In

*Alouatta*, all males and females are trichromats. Colour vision genotypes for all individuals in our SSR study population have been previously reported [5,49–51] but observers collecting data were blind to the monkeys' colour vision genotypes. We did not genotype the Mexican *Alouatta* population but infer them to be trichromats for two reasons: (i) the evolutionary origin of the *Alouatta* L and M opsin genes is inferred to predate the speciation of South American and Mesoamerican *Alouatta* [49] and (ii) the presence of trichromacy has been verified in both South American (e.g. *Alouatta seniculus*) and Mesoamerican (e.g. *A. palliata*) howler monkeys [49,52].

## (e) Data analysis
### (i) Dietary specialization and sensory anatomy
We used Kruskal–Wallis (KW) tests and Dunn *post hoc* tests, implemented in R using the *dunn.test* package [53] to test for significant interspecific variation in diet (mean per cent fruit in annual diet from our literature review) and anatomical measures for which we have data on multiple individuals per species (nasal turbinal surface area, finger ratios and fungiform papillae density). To control for scaling effects of body size, we also calculated 'size-corrected' measures of sensory anatomy. For nasal turbinate surface area, we correct for skull size. Tongue area was not available for the studied species, so we control for body mass when calculating relative tongue papillae density. For rMOB, we correct for brain volume. Because only one MOB value is available for each genus, we do not run statistical tests, but we report the single available aMOB and rMOB values.

### (ii) Foraging behaviour
We restricted behavioural analyses to only include foraging sequences on plant species consumed by all three primate species (electronic supplementary material, table S2). Our final comparative dataset included 26 094 investigation sequences across 2107 feeding bouts (electronic supplementary material table S3; raw data available from the Dryad Digital Repository: https://doi.org/10.5061/dryad.r7sqv9scd). For each feeding bout, we calculated the number of investigation sequences that included (i) sniffing to assess olfaction, (ii) manual touching to assess haptic sense and (iii) bites followed by fruit rejection to assess taste in selection rather than mastication. Due to the high similarity of fig traits, we grouped the nine *Ficus* species into two morpho-groups: i.e. 'conspicuous' (small reddish) figs and 'cryptic' (large, evergreen) figs [15].

To address question 1, we only included data from trichromat individuals ($n = 986$ feeding bouts, 14 328 investigation sequences, electronic supplementary material, table S3) to avoid the confounding influence of polymorphic colour vision on interspecific variation. For each of the sensory behaviours, we performed two generalized linear mixed effects models (GLMMs) with a Poisson distribution for count data: (i) a null model with no fixed effects and (ii) a model including primate species as a fixed effect. For all models, we included the total number of investigation sequences per bout as an offset variable and random effects of focal animal ID and fruit species. For *Alouatta* in SSR, a unique focal ID was assigned to each age/sex class of the social group because monkeys were not individually known. We used likelihood ratio tests (LRTs) to compare model fitness between null and test models. When models that included primate species outperformed the null model, we report the results from GLMMs using incidence rate ratios, which represent the incidence rate of the use of a particular sensory behaviour in test species relative to its incidence rate in a reference species (which we arbitrarily set as *Ateles*) and pairwise contrasts for the fixed effects. The choice of the reference species did not change the results. We calculated incidence rate ratios as the exponent of the fixed effects of the best fit model. All analyses were performed in R v. 3.6.1 using *lme4*, *sjPlot* and *emmeans* packages [54–56].

For question 2, we included data from both trichromat and dichromat individuals of *Cebus* and *Ateles*, and all *Alouatta* individuals. We first categorized fruit species as 'conspicuous' (e.g. colour change during ripening from green to yellow/orange/red) or 'cryptic' (remaining green while ripening, e.g. [29]). Then, for each dependent sensory variable (sniff, touch, bite events) as well as 'acceptance' (fruits eaten/fruits investigated), we performed a GLMM which included the individual's species, individual's colour vision phenotype (trichromat and dichromat) and fruit conspicuity (conspicuous and cryptic) as fixed effects and an interaction between colour vision phenotype and fruit conspicuity, with focal animal ID and fruit species as random effects. Because we included an interaction term and had unequal sample sizes, we designated the largest subclass in each categorical variable as the reference (colour vision phenotype: trichromat, fruit colour: conspicuous) rather than arbitrarily assigning it [57]. We used LRTs to compare these test models to the corresponding null models that only included primate species and fruit conspicuity as fixed effects. To address possible confounding influences of sex and routine versus polymorphic trichromacy on these analyses, we repeated this analysis with only females of polymorphic species (*Ateles* and *Cebus*).

## 3. Results

## (a) Do primate species vary in fruit investigative behaviours reflecting dietary specialization or sensory anatomy?
### (i) Dietary specialization
We found significant differences in fruit consumption between species (KW: $\chi_2^2 = 44.70$, $p < 0.0001$; figure 2*a*; electronic supplementary material, table S4). Average monthly per cent fruit feeding was significantly higher in *Ateles* compared to both *Cebus* and *Alouatta* ($p = 0.010$, $p < 0.0001$, respectively, *post hoc* Dunn's test) and in *Cebus* relative to *Alouatta* ($p = 0.003$). This is consistent with previous classifications of *Ateles* as a ripe fruit specialist, *Alouatta* as the least frugivorous and *Cebus* as intermediate [58]. These results are also consistent with feeding records of these three sympatric species at SSR when they are observed concurrently during the same months and years [58].

### (ii) Sensory anatomy
*Ateles* has the largest aMOB volume (90.4 mm³), while *Cebus* (39.9 mm³) and *Alouatta* (41.4 mm³) do not differ noticeably (electronic supplementary material, table S4). As a per cent of total brain volume, *Ateles* has a slightly higher rMOB value than *Alouatta* (0.09% and 0.08%, respectively), while *Cebus* has a smaller rMOB (0.06%). By contrast, metrics for nasal turbinate surface area suggest a different pattern (KW: $\chi_2^2 = 15.95$, $p < 0.0001$; figure 2*b*; electronic supplementary material, table S4); *Alouatta* has significantly larger surface areas than *Ateles* (*post hoc* Dunn's test; $p = 0.0001$). This result persists even after controlling for differences in skull size (electronic supplementary material, figure S3a, KW: $\chi_2^2 = 16.54$, $p < 0.0001$). Concerning manual dexterity, the thumb : index ratio also differs significantly across the three taxa (KW: $\chi_2^2 = 20.11$, $p < 0.0001$, figure 2*c*; electronic supplementary material, table S4). *Cebus* has a significantly longer thumb relative to the index finger (i.e. a higher ratio)

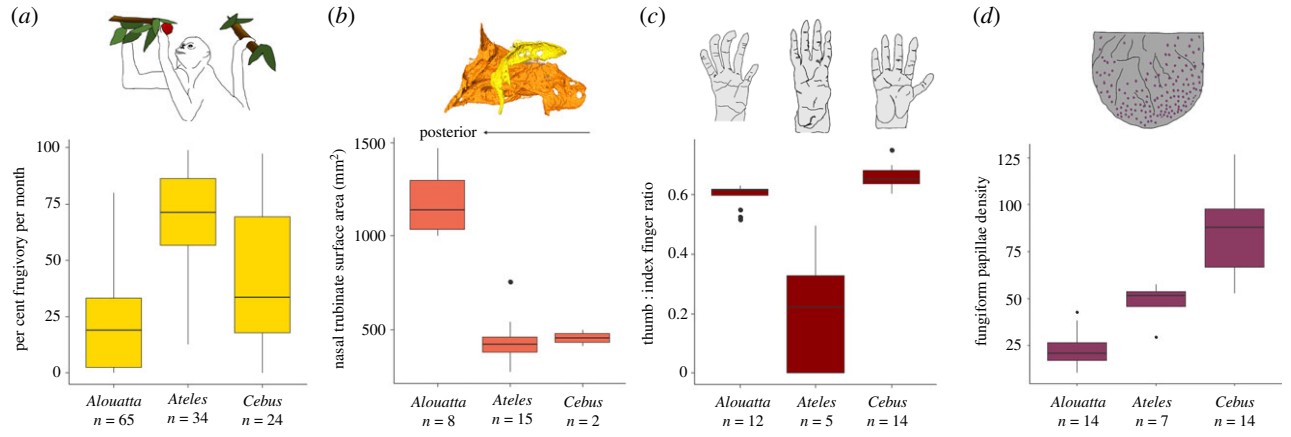

**Figure 2.** Boxplots of interspecific variation for (a) average monthly per cent fruit feeding ($\chi^2_2 = 44.7$, $p < 0.0001$), (b) nasal turbinate surface area ($\chi^2_2 = 15.95$, $p < 0.0001$), (c) thumb : index ratios ($\chi^2_2 = 20.11$, $p < 0.0001$) and (d) density of fungiform papillae ($\chi^2_2 = 23.27$, $p < 0.0001$). Images: (a) *Ateles* feeding on fruit, (b) the reconstructed left nasal turbinate (yellow) and ethmoturbinate (orange) for *Cebus*, (c) sketches of typical *Alouatta*, *Ateles* and *Cebus* hands and (d) fungiform papillae on a tongue. (Online version in colour.)

than *Alouatta* ($p = 0.0049$) and *Ateles* ($p < 0.0001$). *Alouatta* also has relatively longer thumbs compared to *Ateles* ($p = 0.0371$). We also identified significant differences between *Alouatta*, *Ateles* and *Cebus* in the density of fungiform papillae on the anterior tongue (KW: $\chi^2_2 = 23.27$, $p < 0.0001$). *Cebus* has higher papillae densities than *Alouatta* ($p < 0.0001$), but not *Ateles* ($p = 0.098$), while *Ateles* and *Alouatta* do not significantly differ ($p = 0.097$). The pattern remains when body size is controlled for (electronic supplementary material, figure S3b, KW: $\chi^2_2 = 19.86$, $p < 0.0001$), with the additional result that *Cebus* has higher papillae densities than *Ateles* ($p = 0.026$).

### (iii) Foraging behaviour

We observed significant interspecific variation in the proportion of foraging sequences involving olfaction, manual touch and bite followed by reject (figure 3; electronic supplementary material, table S5). Overall, *Ateles* sniffed fruits at 17.8 times the rate of *Cebus* and 5.9 times the rate of *Alouatta*, while *Alouatta* sniffed fruits at three times the rate of *Cebus* (figure 3a). By contrast, *Cebus* used manual touch most often, 1.9 times more often than *Ateles* and 11.8 times more often than *Alouatta*. *Ateles* used manual touch 6.1 times more often than *Alouatta* (figure 3b). *Cebus* and *Alouatta* both had higher frequencies of bite/reject than *Ateles* did (2.9 and 2.2 times, respectively), but did not differ from each other (figure 3c). We detected variation in the use of touch and bite/reject between *Alouatta* from the different sites; however, this variation did not influence the pattern of interspecific differences (electronic supplementary material, results S1), and there was no effect of sex on the use of sensory behaviour in *Alouatta* (electronic supplementary material, results S2).

### (b) Does colour vision phenotype influence the use of sensory behaviours and food selection accuracy?

We identified significant effects of colour vision phenotype and fruit conspicuity on the use of the non-visual senses (table 1; electronic supplementary material, figure S4). Across all three species and both colour vision phenotypes, visually conspicuous fruits were the most common focus of feeding bouts (70–98% of bouts, electronic supplementary material, table S7). Dichromats used sniffing behaviours more frequently relative to trichromats to evaluate fruit,

and this effect was amplified when feeding on cryptic fruit (table 1; electronic supplementary material, figure S4a). While there were no main effects of colour vision or fruit conspicuity on the use of manual touch, there was a significant interaction effect. Specifically, dichromats used touch less often than trichromats when investigating cryptic fruit. Similar to the results for olfaction, dichromats bit and rejected fruits more often than trichromats, particularly when they were feeding on cryptic fruits (table 1; electronic supplementary material, figure S4). We also found a significant interaction between colour vision and fruit conspicuity in how often fruit was 'accepted' (i.e. eaten) during a foraging sequence. While there was no difference in fruit acceptance between dichromats and trichromats for conspicuous fruit, dichromats accepted cryptic fruit at nearly half the rate of trichromats (table 1; electronic supplementary material, figure S4). When we repeated this analysis with only females of polymorphic species (*Ateles* and *Cebus*), we found the same pattern (electronic supplementary material, Results S3, table S6 and figure S4), but also identified significant main effects of fruit conspicuity on the use of manual touch and fruit acceptance. Among *Ateles* and *Cebus* females, cryptic fruits were touched more often and were accepted less often relative to conspicuous fruits.

## 4. Discussion

### (a) Do primate species vary in fruit investigative behaviours reflecting dietary specialization or sensory anatomy?

#### (i) Olfaction

Our results offer mixed support for our hypothesis regarding dietary specialization. As predicted, the highly frugivorous *Ateles* sniffed fruits the most frequently; however, *Cebus* monkeys, the next most frugivorous species, sniffed fruits less frequently than the least frugivorous howler monkeys (*Alouatta*). Importantly, given the limitations of our sample size, and the large dietary flexibility present in *Cebus* [37], which may complicate diet–sensation relationships, more extensive study of a greater number of species would be

**Figure 3.** Incidence rate ratio plots and statistical significance of GLMMs for the effect of primate species on investigation sequences (question 1) including (*a*) sniffing, (*b*) touch and (*c*) bite followed by reject. For each panel, the incidence of foraging sequences using that behaviour for *Alouatta* and *Cebus* are plotted relative to the incidence in *Ateles*. *Ateles* was arbitrarily chosen as the reference species (vertical dashed line at 1). These results include only trichromatic individuals across the three primate species. LRT statistics for each model against the null model and pairwise contrasts were reported for each panel. Photo credits: Carrie Veilleux, Amanda Melin, Fernando Campos. (Online version in colour.)

**Table 1.** Model parameters and likelihood ratio tests for colour vision analyses.

| Behaviour | Test model relative to null model | Primate Species Effect[a] | Colour Vision Effect[b] | Fruit Conspicuity Effect[c] | Fruit Conspicuity × Colour Vision Interaction[d] |
|---|---|---|---|---|---|
| Sniffing | $\chi^2_2 = 68.14^{***}$ | *Alouatta*: $-1.17^{***}$ | $0.49^{*}$ | n.s. | $1.28^{***}$ |
| | | *Cebus*: $-1.32^{***}$ | | | *conspicuous*: $0.49^{*}$ |
| | | | | | *cryptic*: $1.78^{***}$ |
| Manual Touch | $\chi^2_2 = 21.95^{***}$ | *Alouatta*: $-1.68^{***}$ | n.s. | n.s. | $-0.29^{***}$ |
| | | *Cebus*: $0.79^{***}$ | | | *conspicuous*: n.s. |
| | | | | | *cryptic*: $-0.34^{***}$ |
| Bite and Reject | $\chi^2_2 = 28.30^{***}$ | *Alouatta*: $1.33^{***}$ | $0.5^{*}$ | n.s. | $0.99^{**}$ |
| | | *Cebus*: $0.90^{***}$ | | | *conspicuous*: $0.5^{*}$ |
| | | | | | *cryptic*: $1.5^{***}$ |
| Acceptance | $\chi^2_2 = 105.74^{***}$ | *Alouatta*: $0.11^{**}$ | n.s. | n.s. | $-0.63^{***}$ |
| | | *Cebus*: n.s. | | | *conspicuous*: n.s. |
| | | | | | *cryptic*: $-0.60^{***}$ |

[a]Model estimate for *Alouatta* and *Cebus* relative to *Ateles*.
[b]Model estimate for dichromats relative to trichromats.
[c]Model estimate for cryptic relative to conspicuous fruits.
[d]Interaction effect in model, with post-hoc pairwise contrasts depicting effects of dichromats relative to trichromats for conspicuous and cryptic fruits.
Significance: $p > 0.05$ (n.s.), $<0.05$ (*), $<0.01$ (**), $<0.001$ (***).

useful for clarifying the relationships between diet and olfactory specialization.

Regarding olfactory anatomy, our prediction that the volume of the MOB would positively correlate with sniffing behaviour was partially supported: *Ateles* had the largest aMOB and rMOB volume and sniffed fruits the most often. However, while *Alouatta* sniffed fruits more often than *Cebus* and had a slightly larger rMOB, their aMOB volumes were not appreciably different. Interestingly, absolute and relative nasal turbinate surface areas were smallest in *Ateles* (an avid fruit sniffer), and largest for *Alouatta* (intermediate for sniffing behaviours). Our data, although limited to three species, suggest that nasal turbinate volume may not be useful for predicting active sniffing in a fruit foraging context, and might be a better

metric of other aspects of olfaction, an idea that invites future study. Olfactory perception remains a poorly understood area and is influenced by diverse factors, including the shape of the turbinates and nasal cavity, air flow rate across the olfactory epithelium, olfactory neuron density, OR diversity and their responsiveness to natural odours [27,32,59,60]. While we examined a narrow ecological context (e.g. active sniffing by three species in the context of fruit foraging), our data suggest complicated relationships between diet, olfactory structures and behaviour. Future studies integrating additional anatomical and genetic variables, as well as the intensity and breadth of relevant ecological and social stimuli, in a wide diversity of species will further illuminate the relationship between olfactory anatomy and behaviour.

## (ii) Manual touch

We predicted that species more reliant on ripe fruit (i.e. *Ateles*, and to a lesser extent *Cebus*) would use manual touch more often than less frugivorous species, as fruit softening is a reliable cue of ripening for many plant species [16]. Concordant with this, *Ateles* and *Cebus* used manual touch more often than *Alouatta*. Contrary to this prediction, however, the most frugivorous species (*Ateles*) handled fruits less often than *Cebus*, more often eating fruit directly from branches (electronic supplementary material, video S3). We expect this latter result is driven in part by differences in hand morphology. Increased manual dexterity has been associated with higher use of discriminative touch in primates [10]. *Cebus* is known for high manual dexterity [61–63] and is the only platyrrhine observed to use tools [64,65]. They also had the highest thumb : index ratio in our study. The lower use of manual touch in *Ateles* may reflect a trade-off associated with their derived style of suspensory locomotion. Specialized brachiation imposes functional constraints on hand structure and digit proportions [48,66] and *Ateles* in our study either lack an external thumb or possess a small, vestigial remnant, consistent with previous reports [67]. Despite this, the highly frugivorous *Ateles* still touch fruits more often than the more manually dexterous *Alouatta*, which they accomplish through squeezing fruits by gripping them along the length of a bent index finger (electronic supplementary material, figure S4 and video S3). The evolution of this derived manual touch behaviour suggests that palpating fruit is informative to spider monkeys and adaptations favouring discriminative touch may be under selection in frugivorous primates more broadly. This interpretation is consistent with reports of higher mechanoreceptor density in the fingers of frugivorous primates [20].

## (iii) Taste

We did not find support for our predictions regarding diet, gustatory morphology and our metric of tasting behaviour—i.e. non-ingestive biting. *Ateles* performed this behaviour the least frequently, despite being the most frugivorous species. The species that used bite/reject more frequently (*Cebus* and *Alouatta*) had the highest and lowest density of fungiform papillae, respectively, which together suggests that fungiform papillae density may not be a useful metric in predicting where taste is used to reject fruits. However, biting also gives mechanical information about the hardness of the fruits, and species may vary in use of mouth palpating as a mechanosensory cue. Given that our metric (bite/reject) is crude, and the use of taste is also used in fruit acceptance behaviours, the limitations of field research for assessing taste are large at present. Drivers of interspecific variation in this behaviour remain opaque, and study of fruit compounds and taste receptors [68] or behavioural experiments in the same vein as (e.g. [69]) hold considerable promise and would be useful for further exploring this question.

## (b) Does variation in colour vision phenotype influence the use of other sensory behaviours and food selection accuracy?

Members of the present authorship have previously [8,15] reported that dichromatic *Cebus* sniffed fruits significantly more often than trichromats. Our present analyses, which also include data for *Ateles* and *Alouatta*, are consistent with this finding. These results are especially pronounced for cryptic (evergreen) fruit species, which typically produce more volatile compounds and are often dispersed by nocturnal macrosmatic bats [8,70]. We also demonstrate for the first time that dichromatic platyrrhine primates bite and reject fruits more frequently than trichromats. Together, these results provide support for the hypothesis that dichromatic primates rely more heavily on non-visual senses to assess fruit ripeness [15,29]. Surprisingly, we found that trichromats touch cryptic fruits with their hands more often than dichromats, suggesting they use their haptic sense to assess ripeness in the absence of a colour cue. This result is consistent with a recent study on *Alouatta* [35]. As the cryptic fruit species we studied soften upon ripening [71], squeezing fruits may generate a useful non-visual cue that helps with assessing edibility. It is unclear at present why trichromats would rely more heavily than dichromats on this haptic cue and we expand on ideas in our electronic supplementary material. Regardless, we find that variation in one sensory system (colour vision) influences the use of multiple other senses when investigating the same fruit types among three sympatric platyrrhine primates, highlighting the need for future studies exploring multi-modal sensory integration across additional diverse species.

## (c) Sensory behaviours, sensory 'proxies' and ecological importance

While sensory ecology studies often take a broad comparative approach and sample many taxa (e.g. [18,21,33]), we performed a detailed comparison of feeding behaviour records and sensory anatomy for a set of three sympatric species assessing similar natural stimuli. Our results suggest that we need to be thoughtful in interpreting commonly used proxies of sensory reliance. For example, of the primates we examined, we found species that used sniffing behaviours more often had relatively large MOBs, but not relatively large nasal turbinates. Rejection of specific anatomical features as behavioural proxies would be premature. However, our results highlight the importance of recognizing that senses are multifaceted and that different proxies may measure different dimensions of a given sense (e.g. threshold sensitivity versus discrimination versus breadth of stimuli [72]). The relationships between different dimensions within a sense, anatomical proxies, and the roles of that sense during foraging are likely far more complex than often acknowledged. Increased integration of behavioural ecology and comparative anatomy, along with the integration of histology and 'omics data, should help disentangle these complexities and the mechanisms underlying adaptations for different aspects of sensation.

**Ethics.** This research adhered to the laws of Costa Rica, the United States, and Canada and complied with protocols approved by the ACG (R-SINAC-ACG-PI-027-18) (R-025-2014-OT-CONAGEBIO) and by the Canada Research Council for Animal Care through the University of Calgary's Life and Environmental Care Committee (AC19-0167).

**Data accessibility.** The datasets generated and analysed during the current study are available in the Dryad Digital Repository [73] and in the electronic supplementary material datasets [74].

**Authors' contributions.** A.D.M.: conceptualization, data curation, formal analysis, funding acquisition, investigation, methodology, project administration, resources, supervision, writing—original draft and writing—review and editing; C.C.V.: conceptualization, data curation, formal analysis, funding acquisition, methodology, visualization, writing—original draft and writing—review and editing; M.C.J.: data

curation, formal analysis, methodology and writing—review and editing; C.H.: investigation and writing—review and editing; K.G.S.-S.: data curation, investigation and writing—review and editing; I.K.L.: data curation, investigation, methodology, visualization and writing—review and editing; S.E.W.: data curation, investigation and writing—review and editing; R.E.W.: data curation, investigation and writing—review and editing; M.A.M.: data curation, funding acquisition, investigation and writing—review and editing; E.M.-C.: investigation and writing—review and editing; C.M.S.: funding acquisition, project administration, resources, supervision and writing—review and editing; L.H.-S.: conceptualization, funding acquisition, project administration, resources, supervision and writing—review and editing; F.A.: funding acquisition, project administration, resources, supervision and writing—review and editing; S.K.: funding acquisition, project administration, resources, supervision and writing—review and editing.

All authors gave final approval for publication and agreed to be held accountable for the work performed therein.

Conflict of interest declaration. We declare we have no competing interests.

Funding. This research was supported by the Natural Sciences and Engineering Research Council of Canada and the Canada Research Chairs Program (A.D.M.), the University of Calgary (C.C.V., A.D.M. and S.E.W.), the Japan Society for the Promotion of Science 18H04005 (S.K.), 15-11926 (C.H.), the British Academy (C.M.S.), the University of Chester (C.M.S.), Chester Zoo (F.A.), National Geographic Society (F.A.), International Primatological Society (M.A.M.), Animal Behaviour Society (M.A.M.) and Natural Environment Research Council (NE/T000341/1, M.C.J.).

Acknowledgements. We thank R. Blanco Segura and M.M. Chavarria and staff from the Área de Conservación Guanacaste and Ministerio de Ambiente y Energía. Warmest thanks also to A. Guadamuz, A. Blauel, J. Hogan, B. Klug, M. Lemmon, N. Parr, L. Weckman, N. Carrethers and N. Ferrero for field assistance. This research adhered to the laws of Costa Rica, Mexico, the United States, and Canada and complied with protocols approved by the Área de Conservación Guanacaste and by the Canada Research Council for Animal Care through the University of Calgary's Life and Environmental Care Committee.

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
