## [Peer Review File · Proceedings of the Royal Society B: Biological Sciences]

Review History

RSPB-2021-1614.R0 (Original submission)

Review form: Reviewer 1

Recommendation

Major revision is needed (please make suggestions in comments)

Scientific importance: Is the manuscript an original and important contribution to its field?

Good

General interest: Is the paper of sufficient general interest?

Good

Quality of the paper: Is the overall quality of the paper suitable?

Good

Is the length of the paper justified?

Yes

Should the paper be seen by a specialist statistical reviewer?

No

Do you have any concerns about statistical analyses in this paper? If so, please specify them explicitly in your report.

Yes

It is a condition of publication that authors make their supporting data, code and materials available - either as supplementary material or hosted in an external repository. Please rate, if applicable, the supporting data on the following criteria.

Is it accessible?

Yes

Is it clear?

Yes

Is it adequate?

Yes

Do you have any ethical concerns with this paper?

No

Comments to the Author

General comments

Melin et al. investigate sensory foraging within and across three platyrrhine species that differ in dietary specialization and color vision phenotype. The authors compare many different types of datasets to answer their questions on 1) whether there are differences in foraging behaviors (sniffing, touching, biting) among their study species, which differ in dietary specialization, color vision phenotype, and olfactory and thumb/finger ratios, and 2) whether the type of color vision affects the foraging behaviors of interest.

The authors' primary findings are with respect to sniffing (related to degree of frugivory and size of the main olfactory bulb) and to touching fruits with the hands (frugivores, even the thumb-deficient *Ateles*, palpate fruit more than the folivorous *Alouatta*!). Furthermore, MOB is a better predictor for fruit sniffing than nasal turbinate surface area, which leads to an interesting suggestion that these two olfactory morphologies are responsive to different stimuli. With respect to color vision phenotype, the results show that within a species, dichromats sniff (and bite) fruits more than trichromats, and trichromats touch foods more than dichromats.

I appreciated the comparisons among the different types of data, from observational foraging to color vision genotypes to measurements of digits. However, I think the signal of the effects of color vision phenotype on sensory foraging becomes a bit obscured from the interactions of many different types of data within and among three primate taxa.

The comparisons within and among species with respect to color vision phenotypes can be confusing, especially when differences are in contrasting directions depending on whether the comparisons are inter- or intra-specific. For example, the polymorphic *Cebus* and *Ateles* (both more frugivorous than *Alouatta*) touch fruits more than the fully trichromatic and folivorous *Alouatta*, but in comparisons within species, trichromatic individuals touch cryptic fruits more than dichromats (this comparison would not include *Alouatta*). The interaction of color vision phenotype and dietary specialization conflates the signal of specific color vision genotype and sensory foraging variables, especially when one of the species is a folivore and is fully trichromatic.

Since it appears that individual color vision genotypes are known, are there differences in the sensory investigation variables among dichromatic vs trichromatic females within the polymorphic species? Do these parallel the findings of the study as a whole (e.g., more sniffing in the dichromats)? Even if the sample sizes are low for such a comparison, the specificity of the contrast would provide additional confirmation of the larger findings, while alleviating some concerns about potential differences in sex, site, and observation period. Furthermore, are there any differences among the variables of interest between the two *Alouatta* sites?

Given the above, it may make for a “cleaner” signal if *Alouatta* were excluded from the initial analysis.

Specific comments by page [Note: There is a discrepancy between page numbers at the top and bottom right of the page. I am using the page numbers at the top throughout this review.]

Abstract, line 57: should read “...that THE species with the highest...”

p. 6, lines 103-106: While I appreciate that there may be limits on manuscript page numbers, there are some omissions of primary literature, such as Jacobs et al. (1996) as a reference for routine trichromacy in *Alouatta*.

Jacobs GH, Neitz M, Deegan JF II, Neitz J (1996) Trichromatic colour vision in New World monkeys. *Nature* 382:156-158.

p. 8, paragraph starting on line 156: Please provide more details on the focal follows: how often were the groups of each species followed? Were these all day follows? Were females and males followed approximately equally?

p. 8, (b) Dietary specialization: Do the authors’ own data agree with the literature feeding data? There is quite a range in the intraspecific frugivory percentages among the studies cited. The current comparisons are limited to foods that were foraged in common by the three species at both sites (p. 11); however, the large range of values associated with each species in terms of monthly frugivory suggests that there are interannual differences in fruit productivity even within a site that could affect results. As described on pp. 7-8, the study periods in Santa Rosa were largely non-overlapping for the three species, and *Alouatta* was studied at two different sites.

p. 9, Olfaction paragraph: The authors mention that absolute MOB volume is an indicator of olfaction. However, nasal turbinate surface area probably scales with respect to body mass. Did the authors take body mass differences into account when comparing nasal turbinate surface area across species? [Not as relevant to take body mass into account for thumb:index finger measurements since this is a ratio.]

p. 10, Colour vision type paragraph, first sentence: By “well established,” do the authors mean that the color vision genotypes of each individual in their *Cebus* and *Ateles* groups are known?

p. 11, line 241: missing word – “...datasets to include only those plant species in WHICH all three primates species foraged...”

p. 11, lines 251-252: This is a bit confusing. It is not clear what data “...included data from trichromat individuals” is in reference to. It seems to mean (based on later descriptions of models on the next page) data on the total numbers of fruits that were foraged by trichromatic individuals but not that only trichromatic individuals were being compared across the study species? If the latter, then how could dichromats vs trichromats be compared in the GLMM?

p. 11, lines 253-261: Only the results of a likelihood ratio test between a full model and a null are provided in the supplemental materials, but not a summary of the full model. What are the

results for the fixed effects of the full model and the pairwise interactions of their levels (it is fairly standard to report pairwise contrasts with the emmeans package in R)? What are the effects of the random variables?

p. 12, top of page: What is the effort expended (e.g., bouts, time spent) by each of the three species on conspicuous vs cryptic fruits?

p. 13, paragraph on foraging behaviour: references for figures should be for Fig. 3 not 2. This figure is difficult to interpret quickly. It would help if the photos (or just labels) of the monkeys were positioned next to their actual positions on the plot.

This is not related to a specific result, but I ask here since it fits best with questions on foraging: are there sex differences within species with respect to the sensory foraging variables? This would be interesting to show because only females are fully trichromatic within the species that are polymorphic for color vision. Furthermore, are there differences between fully trichromatic and dichromatic females?

p. 13, line 318: Figs. S5 and S6 are misnumbered – there is no Fig. S4.

p. 13, lines 321-323: “Dichromats bit and rejected fruits...2.7 times the rate of trichromats for cryptic fruit...” From Fig. S5, this appears to be true only for Ateles but not Cebus.

p. 16, line 402: should read, “...use of other senses IS intriguing.”

This is not a particularly unexpected finding given that routine trichromatic color vision is negatively associated with olfactory receptor genes (Gilad et al., 2004).
Gilad Y, Wiebe V, Przeworski M, Lancet D, Pääbo S. 2004. Loss of olfactory receptor genes coincides with the acquisition of full trichromatic vision in Primates. *PLoS Biology* 2 (1): 120-125.

Review form: Reviewer 2

Recommendation

Major revision is needed (please make suggestions in comments)

Scientific importance: Is the manuscript an original and important contribution to its field?

Excellent

General interest: Is the paper of sufficient general interest?

Excellent

Quality of the paper: Is the overall quality of the paper suitable?

Excellent

Is the length of the paper justified?

Yes

Should the paper be seen by a specialist statistical reviewer?

No

Do you have any concerns about statistical analyses in this paper? If so, please specify them explicitly in your report.

No

It is a condition of publication that authors make their supporting data, code and materials available - either as supplementary material or hosted in an external repository. Please rate, if applicable, the supporting data on the following criteria.

Is it accessible?

Yes

Is it clear?

Yes

Is it adequate?

Yes

Do you have any ethical concerns with this paper?

No

Comments to the Author

In this study, Melin and collaborators examine the relationship between foraging behavior and sensory anatomy in three species of neotropical monkeys. The study is highly integrative, combining detailed field observations with advanced tools to examine anatomy (e.g., micro-CT scanning). The authors show differences in use of olfaction and touch among species that differ in their degree of frugivory and, interestingly, between dichromatic and trichromatic individuals. Overall, the study is an exciting contribution to the fields of behavioral ecology, functional morphology, and primatology.

The manuscript is well written and straightforward; the study methods and analyses as well as the interpretations of the results seem appropriate. My detailed suggestions and questions are shown below.

L56: the abstract needs an overarching hypothesis.

L71 (and 52-53): this statement needs some refinement about the gaps; e.g., echolocation in bats and cetaceans, and lateral-line sensing have been studied extensively.

L73: animals □ terrestrial frugivores

L97: for those not familiar with primates, it would be useful to add a couple of sentences about the major primate clades (catarrhines, strepsirrhines, platyrrhines) and their vision types. This could even extend to mammals in general, to put into broader context.

L162: I appreciate the inclusion of these videos in the supplement, they are very useful!

L171: clarify if these data were restricted to the study localities, or whole species ranges, and what other parameters (e.g., search engines, terms, years) were used.

L192: clarify which turbinate bones (I see this is in the supplement, but it would be useful to present in the main text); the reference provided is for marmosets – is this extrapolation realistic? Add a caveat if necessary.

L197: should “volumes” here be “areas”? If not, this does not seem appropriate since the area is a better proxy for the olfactory epithelium. Did these analyses also include a size correction? If not, I recommend re-running additional analyses correcting for size (e.g., nasal cavity volume or even body mass) and reporting in the supplement.

L201: justify why 0.5 cm of the tongue was measured.

L204: add which species.

L226: “species” here should be “individuals”, correct?

L240-241: sentence needs editing for clarity.

L259: were these analyses run with different reference species? Do the results change? If so, this needs to be acknowledged and discussed.

L333: these results are promising given the coarse morphological proxies used for olfaction (given the complexity of this system). The authors could acknowledge other additional factors that could potentially blur the relationship between olfactory anatomy and behavior, including that (1) the

shape of the turbinates and nasal cavity affects air flow inside the nose and odorant deposition, (2) the density of olfactory neurons in the OE and expressed ORs may vary. Additionally, could the authors discuss if /how these fruits change in aroma when they are ripe, and how that might relate to behavioral differences? If unknown, this could be mentioned as a future direction.

Decision letter (RSPB-2021-1614.R0)

23-Aug-2021

Dear Dr Melin:

I have now received two reviews and comments from the Associate Editor, and based on these and my own reading of your manuscript am writing to inform you that your manuscript RSPB-2021-1614 entitled "Anatomy and dietary specialization influence sensory behaviour among sympatric frugivorous primates" has, in its current form, been rejected for publication in Proceedings B. However, the reviewers, AE, and I all think that your manuscript has the potential to be important, therefore I am inviting you to submit a revised version of your manuscript with each of the reviewers' comments fully addressed. The reviewers' and AE's comments are appended below, but I encourage you to pay close attention to the comments regarding your analyses, particularly as they relate to the color vision question. I, too, wonder whether it makes more sense to analyze the polymorphic species separately, particularly as you appear to have the genetic information for specific individuals within these populations. Finally, please note that this is not a provisional acceptance, and we will send your revised manuscript back out for review.

Sincerely,
 Dr Sarah Brosnan
 Editor, Proceedings B
 mailto: proceedingsb@royalsociety.org

Associate Editor

Board Member: 1

Comments to Author:

Thank you for submitting your manuscript to Proceedings B. We have now received two reviews of your manuscript. The reviewers agree that the study is interesting, and they appreciated the inclusion of many types of data to answer a broad sensory ecology question. However, they also have several concerns that would need to be addressed before the manuscript could be considered for publication in Proceeding B. Both reviewers would like to see changes to the statistical tests or additional analyses conducted, and one reviewer highlighted areas of the manuscript where clarification is needed to understand how some of the tests were run. In particular, this reviewer is wondering about the effect of including the completely trichromatic species in the same analyses with species that are polymorphic and also whether individual genotypes were known for the polymorphic species (i.e. could individual females of *Cebus* and *Ateles* be identified at trichomats or dichromats? If not, how might this be affecting the results?). Both reviewers also identify areas of the manuscript where the literature needs to be more fully cited and where the methods need more detail. The reviewers provide valuable feedback that will improve a future version of the manuscript.

Reviewer(s)' Comments to Author:

Referee: 1

Comments to the Author(s)

General comments

Melin et al. investigate sensory foraging within and across three platyrrhine species that differ in dietary specialization and color vision phenotype. The authors compare many different types of datasets to answer their questions on 1) whether there are differences in foraging behaviors (sniffing, touching, biting) among their study species, which differ in dietary specialization, color vision phenotype, and olfactory and thumb/finger ratios, and 2) whether the type of color vision affects the foraging behaviors of interest.

The authors' primary findings are with respect to sniffing (related to degree of frugivory and size of the main olfactory bulb) and to touching fruits with the hands (frugivores, even the thumb-deficient *Ateles*, palpate fruit more than the folivorous *Alouatta*!). Furthermore, MOB is a better predictor for fruit sniffing than nasal turbinate surface area, which leads to an interesting suggestion that these two olfactory morphologies are responsive to different stimuli. With respect to color vision phenotype, the results show that within a species, dichromats sniff (and bite) fruits more than trichromats, and trichromats touch foods more than dichromats.

I appreciated the comparisons among the different types of data, from observational foraging to color vision genotypes to measurements of digits. However, I think the signal of the effects of color vision phenotype on sensory foraging becomes a bit obscured from the interactions of many different types of data within and among three primate taxa.

The comparisons within and among species with respect to color vision phenotypes can be confusing, especially when differences are in contrasting directions depending on whether the comparisons are inter- or intra-specific. For example, the polymorphic *Cebus* and *Ateles* (both more frugivorous than *Alouatta*) touch fruits more than the fully trichromatic and folivorous *Alouatta*, but in comparisons within species, trichromatic individuals touch cryptic fruits more than dichromats (this comparison would not include *Alouatta*). The interaction of color vision phenotype and dietary specialization conflates the signal of specific color vision genotype and sensory foraging variables, especially when one of the species is a folivore and is fully trichromatic.

Since it appears that individual color vision genotypes are known, are there differences in the sensory investigation variables among dichromatic vs trichromatic females within the polymorphic species? Do these parallel the findings of the study as a whole (e.g., more sniffing in the dichromats)? Even if the sample sizes are low for such a comparison, the specificity of the

contrast would provide additional confirmation of the larger findings, while alleviating some concerns about potential differences in sex, site, and observation period. Furthermore, are there any differences among the variables of interest between the two *Alouatta* sites? Given the above, it may make for a “cleaner” signal if *Alouatta* were excluded from the initial analysis.

Specific comments by page [Note: There is a discrepancy between page numbers at the top and bottom right of the page. I am using the page numbers at the top throughout this review.]

Abstract, line 57: should read “...that THE species with the highest...”

p. 6, lines 103-106: While I appreciate that there may be limits on manuscript page numbers, there are some omissions of primary literature, such as Jacobs et al. (1996) as a reference for routine trichromacy in *Alouatta*.

Jacobs GH, Neitz M, Deegan JF II, Neitz J (1996) Trichromatic colour vision in New World monkeys. *Nature* 382:156–158.

p. 8, paragraph starting on line 156: Please provide more details on the focal follows: how often were the groups of each species followed? Were these all day follows? Were females and males followed approximately equally?

p. 8, (b) Dietary specialization: Do the authors’ own data agree with the literature feeding data? There is quite a range in the intraspecific frugivory percentages among the studies cited. The current comparisons are limited to foods that were foraged in common by the three species at both sites (p. 11); however, the large range of values associated with each species in terms of monthly frugivory suggests that there are interannual differences in fruit productivity even within a site that could affect results. As described on pp. 7-8, the study periods in Santa Rosa were largely non-overlapping for the three species, and *Alouatta* was studied at two different sites.

p. 9, Olfaction paragraph: The authors mention that absolute MOB volume is an indicator of olfaction. However, nasal turbinate surface area probably scales with respect to body mass. Did the authors take body mass differences into account when comparing nasal turbinate surface area across species? [Not as relevant to take body mass into account for thumb:index finger measurements since this is a ratio.]

p. 10, Colour vision type paragraph, first sentence: By “well established,” do the authors mean that the color vision genotypes of each individual in their *Cebus* and *Ateles* groups are known?

p. 11, line 241: missing word – “...datasets to include only those plant species in WHICH all three primates species foraged...”

p. 11, lines 251-252: This is a bit confusing. It is not clear what data “...included data from trichromat individuals” is in reference to. It seems to mean (based on later descriptions of models on the next page) data on the total numbers of fruits that were foraged by trichromatic individuals but not that only trichromatic individuals were being compared across the study species? If the latter, then how could dichromats vs trichromats be compared in the GLMM?

p. 11, lines 253-261: Only the results of a likelihood ratio test between a full model and a null are provided in the supplemental materials, but not a summary of the full model. What are the results for the fixed effects of the full model and the pairwise interactions of their levels (it is fairly standard to report pairwise contrasts with the emmeans package in R)? What are the effects of the random variables?

p. 12, top of page: What is the effort expended (e.g., bouts, time spent) by each of the three species on conspicuous vs cryptic fruits?

p. 13, paragraph on foraging behaviour: references for figures should be for Fig. 3 not 2. This figure is difficult to interpret quickly. It would help if the photos (or just labels) of the monkeys were positioned next to their actual positions on the plot.

This is not related to a specific result, but I ask here since it fits best with questions on foraging: are there sex differences within species with respect to the sensory foraging variables? This would be interesting to show because only females are fully trichromatic within the species that are polymorphic for color vision. Furthermore, are there differences between fully trichromatic and dichromatic females?

p. 13, line 318: Figs. S5 and S6 are misnumbered – there is no Fig. S4.

p. 13, lines 321-323: “Dichromats bit and rejected fruits...2.7 times the rate of trichromats for cryptic fruit...” From Fig. S5, this appears to be true only for *Ateles* but not *Cebus*.

p. 16, line 402: should read, “...use of other senses IS intriguing.”

This is not a particularly unexpected finding given that routine trichromatic color vision is negatively associated with olfactory receptor genes (Gilad et al., 2004).

Gilad Y, Wiebe V, Przeworski M, Lancet D, Pääbo S. 2004. Loss of olfactory receptor genes coincides with the acquisition of full trichromatic vision in Primates. *PLoS Biology* 2 (1): 120-125.

Referee: 2

Comments to the Author(s)

In this study, Melin and collaborators examine the relationship between foraging behavior and sensory anatomy in three species of neotropical monkeys. The study is highly integrative, combining detailed field observations with advanced tools to examine anatomy (e.g., micro-CT scanning). The authors show differences in use of olfaction and touch among species that differ in their degree of frugivory and, interestingly, between dichromatic and trichromatic individuals. Overall, the study is an exciting contribution to the fields of behavioral ecology, functional morphology, and primatology.

The manuscript is well written and straightforward; the study methods and analyses as well as the interpretations of the results seem appropriate. My detailed suggestions and questions are shown below.

L56: the abstract needs an overarching hypothesis.

L71 (and 52-53): this statement needs some refinement about the gaps; e.g., echolocation in bats and cetaceans, and lateral-line sensing have been studied extensively.

L73: animals □ terrestrial frugivores

L97: for those not familiar with primates, it would be useful to add a couple of sentences about the major primate clades (catarrhines, strepsirrhines, platyrrhines) and their vision types. This could even extend to mammals in general, to put into broader context.

L162: I appreciate the inclusion of these videos in the supplement, they are very useful!

L171: clarify if these data were restricted to the study localities, or whole species ranges, and what other parameters (e.g., search engines, terms, years) were used.

L192: clarify which turbinate bones (I see this is in the supplement, but it would be useful to present in the main text); the reference provided is for marmosets – is this extrapolation realistic? Add a caveat if necessary.

L197: should “volumes” here be “areas”? If not, this does not seem appropriate since the area is a better proxy for the olfactory epithelium. Did these analyses also include a size correction? If not, I recommend re-running additional analyses correcting for size (e.g., nasal cavity volume or even body mass) and reporting in the supplement.

L201: justify why 0.5 cm of the tongue was measured.

L204: add which species.

L226: “species” here should be “individuals”, correct?

L240-241: sentence needs editing for clarity.

L259: were these analyses run with different reference species? Do the results change? If so, this needs to be acknowledged and discussed.

L333: these results are promising given the coarse morphological proxies used for olfaction (given the complexity of this system). The authors could acknowledge other additional factors that could potentially blur the relationship between olfactory anatomy and behavior, including that (1) the shape of the turbinates and nasal cavity affects air flow inside the nose and odorant deposition, (2) the density of olfactory neurons in the OE and expressed ORs may vary.

Additionally, could the authors discuss if /how these fruits change in aroma when they are ripe, and how that might relate to behavioral differences? If unknown, this could be mentioned as a future direction.

Author's Response to Decision Letter for (RSPB-2021-1614.R0)

See Appendix A.

RSPB-2022-0048.R0

Review form: Reviewer 1

Recommendation

Accept with minor revision (please list in comments)

Scientific importance: Is the manuscript an original and important contribution to its field?

Excellent

General interest: Is the paper of sufficient general interest?

Excellent

Quality of the paper: Is the overall quality of the paper suitable?

Good

Is the length of the paper justified?

Yes

Should the paper be seen by a specialist statistical reviewer?

No

Do you have any concerns about statistical analyses in this paper? If so, please specify them explicitly in your report.

Yes

It is a condition of publication that authors make their supporting data, code and materials available - either as supplementary material or hosted in an external repository. Please rate, if applicable, the supporting data on the following criteria.

Is it accessible?

Yes

Is it clear?

Yes

Is it adequate?

Yes

Do you have any ethical concerns with this paper?

No

Comments to the Author

General comments

Melin et al. investigate sensory foraging within and across three platyrrhine species that differ in dietary specialization and colour vision phenotype. The authors pose two main questions: 1) are there differences in foraging behaviours (sniffing, touching, biting) among their study species related to degree of frugivory and sensory anatomy (olfactory bulb volumes and turbinate surface areas, fungiform papillae density, and thumb/finger ratios)?, and 2) is colour vision phenotype related to the foraging behaviours of interest and the accuracy of food selection? These questions are compared using two datasets: trichromatic individuals of all three species for the first question and all individuals for the second. In this revision, the authors have performed additional analyses (in the supplementary material) that compared the howler monkeys from the two different sites (for Question 1) and compared the females in the species polymorphic for colour vision for Question 2.

The authors' primary findings are that the most frugivorous species (*Ateles*) sniffs fruit more and the most manually dexterous species (*Cebus*) touches fruit the most. Main olfactory bulb volume is a better predictor for fruit sniffing than nasal turbinate surface area, which leads to an interesting suggestion that these two olfactory morphologies are responsive to different stimuli. With respect to colour vision phenotype, the results show that across all three species and for females within the polymorphic species, dichromats sniff and bite fruits more than trichromats, and trichromats touch and accept cryptic fruits more than dichromats.

The manuscript is well-written and detailed and clearly lays out the many datasets that were used in the analyses. I have to applaud the tremendous effort by the authors to add supplementary results to reinforce their initial conclusions. In particular, the authors have added a supplemental analysis comparing dichromatic and trichromatic females within the polymorphic species that show that the signal is similar to (or in the same direction as) the findings of the larger dataset. The new analysis (Fig. S3, Table S6), in which the authors excluded *Alouatta*, have addressed my prior concerns and have reinforced the results of the larger dataset on how colour vision phenotype interacts with the behavioural variables measured. An added benefit of analysing the polymorphic species is that the comparisons are confined to the frugivorous taxa. The numbers of different types of data (foraging behavioural traits, sensory anatomy, colour vision phenotypes) compared and integrated here will make this a valuable contribution to the field and a good jumping off point for further research.

Specific comments by line

Title: "frugivorous primates" suggests that the taxa studied are all frugivores. I don't think *Alouatta* would be considered one...

Abstract, line 54: "We find the most frugivorous species sniffs fruits the most often..." While this is not wrong, it is a bit misleading since most of these foraging behaviour variables are not well associated with degree of frugivory. In terms of sniffing, the pattern is related to the relative size of the MOB in the three species. Perhaps this could be rephrased?

Line 138: reference to Fig. 1 for study species is no longer applicable.

Line 232: I asked previously about the effort expended (e.g., bouts, time spent) by each of the three species on conspicuous vs cryptic fruits. I agree that the overall percentages of conspicuous vs cryptic fruits in the diet are not germane to the question of fruit ripening. However,

understanding the relative contributions to the diet of these fruits would affect how one would interpret the importance of associated behaviours. For example, even if the monkeys used manual touch more frequently on cryptic foods and rejected them more often (at least among the dichromats), if cryptic foods are not eaten that frequently to begin with, is this really that relevant? Could the authors provide sample sizes in the ms. for conspicuous and cryptic fruits used in their analysis to get a sense of the relative contributions of these foods to the diet? (I realise the full dataset is in the Dryad repository.)

Line 234: missing word: "...we only included data from trichromatic individuals...to avoid THE confounding influence..."

Line 267: dietary specialization results. I asked in the prior version of this ms. about the literature feeding data. In the Supplementary Datasets that the authors nicely provided, frugivory percentages for the same month vary widely between studies of the Cebus monkeys (this does not appear to be such an issue with Alouatta or Ateles). Given this high variance, it may not be that surprising that the behavioural variables examined were not well-associated with degree of frugivory.

Line 268: Though the results of the statistical tests for all the results shown in Fig. 1 are given in the text, could these be included in the figure caption or added to Supplementary Table S4?

Line 300: I asked previously if there were any differences among the variables of interest between the two Alouatta sites? The authors have added a new analysis to address this (Supplementary results S1). The howler monkeys at the two sites differed considerably in touch and biting/fruit reject. When compared to the other taxa in the larger dataset, either individually or combined, however, the authors found no differences in the results. Again, this gives greater confidence in the overall interpretation of the larger dataset, and I appreciate the additional effort made for this comparison.

Line 366: this is supposed to be supplementary Fig S4? This figure is not strictly necessary, but if it is included, then a figure of either a capuchin or howler monkey should also be added as a comparison.

Figures:

Fig. 1: Please add results of statistical tests to caption.

Fig. 2: Thank you for repositioning the species photos next to their plot positions – it makes it easier to read. Perhaps add to the caption that the dataset is for Question 1 with trichromatic individuals of all three species in the analysis.

Fig. S3: Authors changed plot types from incident rate ratios to marginal effects plots of interactions. This new figure is much easier to interpret.

Fig. S4: not referenced in manuscript, but also not strictly necessary.

Other:

Results S4: not referenced in manuscript, but also not strictly necessary.

Review form: Reviewer 3

Recommendation

Reject – article is not of sufficient interest (we will consider a transfer to another journal)

Scientific importance: Is the manuscript an original and important contribution to its field?

Acceptable

General interest: Is the paper of sufficient general interest?

Acceptable

Quality of the paper: Is the overall quality of the paper suitable?

Marginal

Is the length of the paper justified?

Yes

Should the paper be seen by a specialist statistical reviewer?

No

Do you have any concerns about statistical analyses in this paper? If so, please specify them explicitly in your report.

No

It is a condition of publication that authors make their supporting data, code and materials available - either as supplementary material or hosted in an external repository. Please rate, if applicable, the supporting data on the following criteria.

Is it accessible?

Yes

Is it clear?

Yes

Is it adequate?

Yes

Do you have any ethical concerns with this paper?

No

Comments to the Author

Please see attached review (Appendix B).

Decision letter (RSPB-2022-0048.R0)

08-Mar-2022

We have now received referees' reports on your revised manuscript RSPB-2022-0048 entitled "Anatomy and dietary specialization influence sensory behaviour among sympatric frugivorous primates". This was reviewed by one of the original referees and also by a new referee.

We are all agreed that this is a very interesting topic, and that the revised manuscript is much improved. However both referees of this version still have substantial concerns which will require major revision. I am therefore rejecting the manuscript in its current form, but inviting a resubmission, provided the comments of the referees are fully addressed. We do not usually allow a second round of major revisions, but will be willing to consider this if you can address all points. Please note in particular the need to acknowledge the limitations of the study in terms of the general conclusions that can be drawn. I must also emphasise that this is not a provisional acceptance.

- 1) A 'response to referees' document including details of how you have responded to the comments, and the adjustments you have made.
- 2) A clean copy of the manuscript and one with 'tracked changes' indicating your 'response to referees' comments document.
- 3) Line numbers in your main document.
- 4) Please read our data sharing policies to ensure that you meet our requirements <https://royalsociety.org/journals/authors/author-guidelines/#data>.

Yours sincerely,
 Professor Loeske Kruuk
 Editor
 mailto: proceedingsb@royalsociety.org

Associate Editor Board Member
 Comments to Author:

Thank you for resubmitting your manuscript to Proceedings B. We have now received two reviews of your manuscript, one of which is from a reviewer of your initial submission. Both reviewers agree that this study addresses a question that is important and rarely investigated in wild primates. They also greatly appreciate the enormous effort that went into obtaining these data. Unfortunately, the reviewers still raise issues that will require substantial revisions to address. Both reviewers find that some of the conclusions in the manuscript are stated more strongly than can be supported by the results. One of the reviewers had concerns about the interpretation of the results of some cited literature, specifically how active sniffing and passive smelling are processed in the brain. This reviewer was also concerned about the justification for using certain measurements, e.g. absolute vs. relative MOB size and turbinate surface area without knowing the proportion of that area covered in olfactory epithelium. The mix of supported and unsupported predictions could be related to the choices for measurements included in the analyses. I hope the feedback from the reviewers is helpful to you for a future version of the manuscript.

Reviewer(s)' Comments to Author:

Referee: 1

Comments to the Author(s).

General comments

Melin et al. investigate sensory foraging within and across three platyrrhine species that differ in dietary specialization and colour vision phenotype. The authors pose two main questions: 1) are there differences in foraging behaviours (sniffing, touching, biting) among their study species related to degree of frugivory and sensory anatomy (olfactory bulb volumes and turbinate surface areas, fungiform papillae density, and thumb/finger ratios)?, and 2) is colour vision phenotype related to the foraging behaviours of interest and the accuracy of food selection? These questions are compared using two datasets: trichromatic individuals of all three species for the

first question and all individuals for the second. In this revision, the authors have performed additional analyses (in the supplementary material) that compared the howler monkeys from the two different sites (for Question 1) and compared the females in the species polymorphic for colour vision for Question 2.

The authors' primary findings are that the most frugivorous species (*Ateles*) sniffs fruit more and the most manually dexterous species (*Cebus*) touches fruit the most. Main olfactory bulb volume is a better predictor for fruit sniffing than nasal turbinate surface area, which leads to an interesting suggestion that these two olfactory morphologies are responsive to different stimuli. With respect to colour vision phenotype, the results show that across all three species and for females within the polymorphic species, dichromats sniff and bite fruits more than trichromats, and trichromats touch and accept cryptic fruits more than dichromats.

The manuscript is well-written and detailed and clearly lays out the many datasets that were used in the analyses. I have to applaud the tremendous effort by the authors to add supplementary results to reinforce their initial conclusions. In particular, the authors have added a supplemental analysis comparing dichromatic and trichromatic females within the polymorphic species that show that the signal is similar to (or in the same direction as) the findings of the larger dataset. The new analysis (Fig. S3, Table S6), in which the authors excluded *Alouatta*, have addressed my prior concerns and have reinforced the results of the larger dataset on how colour vision phenotype interacts with the behavioural variables measured. An added benefit of analysing the polymorphic species is that the comparisons are confined to the frugivorous taxa. The numbers of different types of data (foraging behavioural traits, sensory anatomy, colour vision phenotypes) compared and integrated here will make this a valuable contribution to the field and a good jumping off point for further research.

Specific comments by line

Title: "frugivorous primates" suggests that the taxa studied are all frugivores. I don't think *Alouatta* would be considered one...

Abstract, line 54: "We find the most frugivorous species sniffs fruits the most often..." While this is not wrong, it is a bit misleading since most of these foraging behaviour variables are not well associated with degree of frugivory. In terms of sniffing, the pattern is related to the relative size of the MOB in the three species. Perhaps this could be rephrased?

Line 138: reference to Fig. 1 for study species is no longer applicable.

Line 232: I asked previously about the effort expended (e.g., bouts, time spent) by each of the three species on conspicuous vs cryptic fruits. I agree that the overall percentages of conspicuous vs cryptic fruits in the diet are not germane to the question of fruit ripening. However, understanding the relative contributions to the diet of these fruits would affect how one would interpret the importance of associated behaviours. For example, even if the monkeys used manual touch more frequently on cryptic foods and rejected them more often (at least among the dichromats), if cryptic foods are not eaten that frequently to begin with, is this really that relevant? Could the authors provide sample sizes in the ms. for conspicuous and cryptic fruits used in their analysis to get at a sense of the relative contributions of these foods to the diet? (I realise the full dataset is in the Dryad repository.)

Line 234: missing word: "...we only included data from trichromatic individuals...to avoid THE confounding influence..."

Line 267: dietary specialization results. I asked in the prior version of this ms. about the literature feeding data. In the Supplementary Datasets that the authors nicely provided, frugivory percentages for the same month vary widely between studies of the *Cebus* monkeys (this does not appear to be such an issue with *Alouatta* or *Ateles*). Given this high variance, it may not be

that surprising that the behavioural variables examined were not well-associated with degree of frugivory.

Line 268: Though the results of the statistical tests for all the results shown in Fig. 1 are given in the text, could these be included in the figure caption or added to Supplementary Table S4?

Line 300: I asked previously if there were any differences among the variables of interest between the two *Alouatta* sites? The authors have added a new analysis to address this (Supplementary results S1). The howler monkeys at the two sites differed considerably in touch and biting/fruit reject. When compared to the other taxa in the larger dataset, either individually or combined, however, the authors found no differences in the results. Again, this gives greater confidence in the overall interpretation of the larger dataset, and I appreciate the additional effort made for this comparison.

Line 366: this is supposed to be supplementary Fig S4? This figure is not strictly necessary, but if it is included, then a figure of either a capuchin or howler monkey should also be added as a comparison.

Figures:

Fig. 1: Please add results of statistical tests to caption.

Fig. 2: Thank you for repositioning the species photos next to their plot positions – it makes it easier to read. Perhaps add to the caption that the dataset is for Question 1 with trichromatic individuals of all three species in the analysis.

Fig. S3: Authors changed plot types from incident rate ratios to marginal effects plots of interactions. This new figure is much easier to interpret.

Fig. S4: not referenced in manuscript, but also not strictly necessary.

Other:

Results S4: not referenced in manuscript, but also not strictly necessary.

Referee: 3

Comments to the Author(s).

Please see attached review (Appendix B).

Author's Response to Decision Letter for (RSPB-2022-0048.R0)

See Appendix C.

RSPB-2022-0847.R0

Review form: Reviewer 3

Recommendation

Accept with minor revision (please list in comments)

Scientific importance: Is the manuscript an original and important contribution to its field?

Good

General interest: Is the paper of sufficient general interest?

Good

Quality of the paper: Is the overall quality of the paper suitable?

Acceptable

Is the length of the paper justified?

Yes

Should the paper be seen by a specialist statistical reviewer?

No

Do you have any concerns about statistical analyses in this paper? If so, please specify them explicitly in your report.

No

It is a condition of publication that authors make their supporting data, code and materials available - either as supplementary material or hosted in an external repository. Please rate, if applicable, the supporting data on the following criteria.

Is it accessible?

Yes

Is it clear?

Yes

Is it adequate?

Yes

Do you have any ethical concerns with this paper?

No

Comments to the Author

I appreciate the revised version of the manuscript that the authors submitted, and I find it to be very strong still in a number of areas. I think it approaches the field of comparative sensory biology from a novel and robust perspective, and for this reason it makes a valuable contribution.

My final remaining comment still concerns how to interpret these results in the broader context of the field. For examples, lines 350-351 and 377-379 are still a bit heavy-handed, in my opinion. But, overall I think this is a very minor critique and I am happy with how the authors have contextualized their findings and offered suggestions going forward.

Decision letter (RSPB-2022-0847.R0)

24-Jun-2022

Dear Dr Melin

I am pleased to inform you that your manuscript RSPB-2022-0847 entitled "Anatomy and dietary specialization influence sensory behaviour among sympatric frugivorous primates" has been accepted for publication in Proceedings B. My apologies for the delay in returning a decision to you. The reviewer and Associate Editor's comments (not including confidential comments to the

Editor) are included at the end of this email for your reference, as well as a couple of minor ones from me.

The paper is much improved, but there is still some concern about its limitations given the small number of species used. The Associate Editor who reviewed this version has also made some very important points with regard to the results being clear for a non-specialist (clarifying points that I was certainly finding puzzling). We would therefore like to invite you to revise your manuscript to address these issues. Toning down any general conclusions regarding different types of species is essential. In many cases, this can be done very simply by making it clear that you are just referring to the species considered here (e.g. L362, change 'We predicted that species more reliant on ripe fruit would...' to 'We predicted that the species more reliant on ripe fruit (species X and species Y) would...'). Please check through carefully for any other such instances, and please address the AE's comment below about acknowledging the limitations of only using three closely-related species.

Because the schedule for publication is very tight, it is a condition of publication that you submit the revised version of your manuscript within 7 days. If you do not think you will be able to meet this date please let us know.

When submitting your revision please upload a file under "Response to Referees" - in the "File Upload" section. This should document, point by point, how you have responded to the reviewers' and Editors' comments, and the adjustments you have made to the manuscript. We also require a copy of the revised manuscript showing track changes to be uploaded.

- 4) Data accessibility section and data citation

It is a condition of publication that data supporting your paper are made available either in the electronic supplementary material. Authors must complete the 'data accessibility' section in the submission system. This should list the database and accession number for all data from the article that has been made publicly available, for instance:

NB. From April 1 2013, peer reviewed articles based on research funded wholly or partly by RCUK must include, if applicable, a statement on how the underlying research materials – such as data, samples or models – can be accessed.

[http://datadryad.org/submit?journalID=RSPB&manu=\(Document not available\)](http://datadryad.org/submit?journalID=RSPB&manu=(Document%20not%20available)) which will take you to your unique entry in the Dryad repository. If you have already submitted your data to dryad you can make any necessary revisions to your dataset by following the above link.

Please include the Dryad DOI in the Data Accessibility section and reference in the paper's bibliography.

Please see our Data Sharing Policies (<https://royalsociety.org/journals/authors/author-guidelines/>).

6) A media summary: a short non-technical summary (up to 100 words) of the key findings/importance of your manuscript.

Once again, thank you for submitting your manuscript to *Proceedings B* and I look forward to receiving your revision. If you have any questions at all, please do not hesitate to get in touch.

Sincerely,
Professor Loeske Kruuk
mailto: proceedingsb@royalsociety.org

Associate Editor
Board Member: 1
Comments to Author:

I am seeing this paper for the first time as a reviewer. In general, I like it quite a lot; its a neat system and I am pleased to see the authors comparing three sympatric species. I have just a couple of big picture comments that are needed to make this manuscript ready for publication.

First, while this is a fantastic system, at times the authors are overstretching their conclusions; it is only three species and more than that, it's three very closely related platyrrhines. How do we know that these results will generalize to the rest of the primates, much less other mammals or animals? A bit more circumspection is needed when interpreting these results.

Second, quite a lot of the paper is written as if for a specialist audience, which means that some of the most interesting components are not going to be evident to the average reader. For instance, nowhere in the abstract do you mention the unusual color vision system of the Cebus and Ateles monkeys, which makes them a particularly good pair of species to study as you can compare di- and tri-chromats within-species. This oversight is curious and seems to assume that the reader is sufficiently familiar with this system that it would be obvious. Please read over your paper carefully to ensure that it is written so that you neither overstep the limits of your data nor make

unwarranted assumptions about what your reader already knows; after this is done I think this will be a nice addition to the literature.

Editor (LK) minor comments

Where you cite a statistic, be sure to include the relevant d.f., e.g. for the chi-squared values at L274.

Typo at end of L289.

L338 - check syntax. (Consider rewriting this sentence, as it is not easy to follow.)

Reviewer(s)' Comments to Author:

Referee: 3

Comments to the Author(s).

I appreciate the revised version of the manuscript that the authors submitted, and I find it to be very strong still in a number of areas. I think it approaches the field of comparative sensory biology from a novel and robust perspective, and for this reason it makes a valuable contribution.

My final remaining comment still concerns how to interpret these results in the broader context of the field. For examples, lines 350-351 and 377-379 are still a bit heavy-handed, in my opinion. But, overall I think this is a very minor critique and I am happy with how the authors have contextualized their findings and offered suggestions going forward.

Author's Response to Decision Letter for (RSPB-2022-0847.R0)

See Appendix D.

Decision letter (RSPB-2022-0847.R1)

18-Jul-2022

Dear Dr Melin,

Thanks for submitting a revised version of this manuscript, and for the the changes you have made to address the previous comments.

I was hoping to be able to accept this version without any further changes. However, I'm afraid that the Abstract is still not addressing the problem of explaining the system clearly enough to readers who do not already know the details. In particular, it is not clear in the Abstract that capuchins and spider monkeys are both di- and tri-chromats. I appreciate that you say at the start of the Abstract that you use within-species variation on a range of traits, and that you mention 'controlling for species', but the abstract needs to say explicitly that there is a polymorphism in vision type within species in your study species. Please refer back to the explicit comments on the previous version from the AE about not having mentioned the polymorphism in the abstract: 'This oversight is curious and seems to assume that the reader is sufficiently familiar with this system that it would be obvious'.

Please therefore revise your manuscript to address this important issue with the abstract (I suggest giving it to non-experts in the field to test if the polymorphism aspect is clear to them.) Because the schedule for publication is very tight, it is a condition of publication that you submit

the revised version of your manuscript within 7 days. If you do not think you will be able to meet this date please let us know.

NB. From April 1 2013, peer reviewed articles based on research funded wholly or partly by RCUK must include, if applicable, a statement on how the underlying research materials – such

as data, samples or models – can be accessed. This statement should be included in the data accessibility section.

[http://datadryad.org/submit?journalID=RSPB&manu=\(Document not available\)](http://datadryad.org/submit?journalID=RSPB&manu=(Document+not+available)) which will take you to your unique entry in the Dryad repository. If you have already submitted your data to dryad you can make any necessary revisions to your dataset by following the above link. Please see <https://royalsociety.org/journals/ethics-policies/data-sharing-mining/> for more details.

Sincerely,
Professor Loeske Kruuk
Editor, Proceedings B
<mailto:proceedingsb@royalsociety.org>

Author's Response to Decision Letter for (RSPB-2022-0847.R1)

See Appendix E.

Decision letter (RSPB-2022-0847.R2)

25-Jul-2022

Dear Dr Melin

I am pleased to inform you that your manuscript entitled "Anatomy and dietary specialization influence sensory behaviour among sympatric frugivorous primates" has been accepted for publication in Proceedings B.

Data Accessibility section

Open Access

You are invited to opt for Open Access, making your freely available to all as soon as it is ready for publication under a CCBY licence. Our article processing charge for Open Access is £1700. For more information please visit <http://royalsocietypublishing.org/open-access>.

If you have opted for Open Access in Proceedings B, payment of an article processing charge (APC) may be due before your article is published. Our partner Copyright Clearance Center's RightsLink for Scientific Communications will contact the corresponding author about your open access options from the email domain @copyright.com (if you have any queries regarding fees, please see <https://royalsocietypublishing.org/rspb/for-authors#question12> or contact authorfees@royalsociety.org).

Paper charges

Sincerely,

Proceedings B

Appendix A

23-Aug-2021

Dear Dr Melin:

I have now received two reviews and comments from the Associate Editor, and based on these and my own reading of your manuscript am writing to inform you that your manuscript RSPB-2021-1614 entitled "Anatomy and dietary specialization influence sensory behaviour among sympatric frugivorous primates" has, in its current form, been rejected for publication in Proceedings B. However, the reviewers, AE, and I all think that your manuscript has the potential to be important, therefore I am inviting you to submit a revised version of your manuscript with each of the reviewers' comments fully addressed. The reviewers' and AE's comments are appended below, but I encourage you to pay close attention to the comments regarding your analyses, particularly as they relate to the color vision question. I, too, wonder whether it makes more sense to analyze the polymorphic species separately, particularly as you appear to have the genetic information for specific individuals within these populations. Finally, please note that this is not a provisional acceptance, and we will send your revised manuscript back out for review.

We are grateful to Editor Brosnan and the AE for their thoughtful comments and for the opportunity to submit a revised version of our manuscript. Below we provide detailed responses to the suggestions provided by the AE and reviewers, including revised analyses. Line numbers refer to the main text. We also provide a document with all changes tracked. Finally, we have made minor edits to our text throughout and removed a table and a figure to make room for the additional analyses and expansions requested. We hope our manuscript is now found suitable for publication in Proceedings B.

Sincerely,

Dr Sarah Brosnan
Editor, Proceedings B
mailto: proceedingsb@royalsociety.org

Associate Editor
Board Member: 1
Comments to Author:

Thank you for submitting your manuscript to Proceedings B. We have now received two reviews of your manuscript. The reviewers agree that the study is interesting, and they appreciated the inclusion of many types of data to answer a broad sensory ecology question. However, they also have several concerns that would need to be addressed before the manuscript could be considered for publication in Proceeding B. Both reviewers would like to see changes to the statistical tests or additional analyses conducted, and one reviewer highlighted areas of the manuscript where clarification is needed to understand how some of the tests were run. In particular, this reviewer is wondering about the effect of including the completely trichromatic species in the same analyses with species that are polymorphic and also whether individual genotypes were known for the polymorphic species (i.e. could individual females of *Cebus* and *Ateles* be identified at trichomats or dichromats? If not, how might this be affecting the results?). Both reviewers also identify areas of the manuscript where the literature needs to be more fully cited and where the methods need more detail. The reviewers provide valuable feedback that will improve a future version of the manuscript.

We appreciate the comments from the Associate Editor. In response to the numerous helpful suggestions and detailed comments from the reviewers, we have both revised and clarified our analyses to address each of these points, which we detail below. In addition, we have addressed the requests for revisions to the writing, including citing additional literature and revising the methods section. We removed a figure showing our study species and removed a table that overlapped with a data figure in order to make room for new analyses and text. We feel our manuscript is much improved and hope that is now found suitable for publication in Proceedings B.

Reviewer(s)' Comments to Author:

Referee: 1

Comments to the Author(s)
General comments

Melin et al. investigate sensory foraging within and across three platyrrhine species that differ in dietary specialization and color vision phenotype. The authors compare many different types of datasets to answer their questions on 1) whether there are differences in foraging behaviors (sniffing, touching, biting) among their study species, which differ in dietary specialization, color vision phenotype, and olfactory and thumb/finger ratios, and 2) whether the type of color vision affects the foraging behaviors of interest.

The authors' primary findings are with respect to sniffing (related to degree of frugivory and size of the main olfactory bulb) and to touching fruits with the hands (frugivores, even the thumb-deficient *Ateles*, palpate fruit more than the folivorous *Alouatta*!). Furthermore, MOB is a better predictor for fruit sniffing than nasal turbinate surface area, which leads to an interesting suggestion that these two olfactory morphologies are responsive to different stimuli. With respect to color vision phenotype, the results show that within a species, dichromats sniff (and bite) fruits more than trichromats, and trichromats touch foods more than dichromats.

1.1. I appreciated the comparisons among the different types of data, from observational foraging to color vision genotypes to measurements of digits. However, I think the signal of the effects of color vision phenotype on sensory foraging becomes a bit obscured from the interactions of many different types of data within and among three primate taxa.

We are pleased the reviewer appreciated the integrative approach we pursued. We are grateful for the many useful suggestions that allowed us to improve our manuscript and we have revised our text and included new analyses to better present the color vision results. Below we address each point in detail.

1.2. The comparisons within and among species with respect to color vision phenotypes can be confusing, especially when differences are in contrasting directions depending on whether the comparisons are inter- or intra-specific. For example, the polymorphic *Cebus* and *Ateles* (both more frugivorous than *Alouatta*) touch fruits more than the fully trichromatic and folivorous *Alouatta*, but in comparisons within species, trichromatic individuals touch cryptic fruits more than dichromats (this comparison would not include *Alouatta*). The interaction of color vision phenotype and dietary specialization conflates the signal of specific color vision genotype and sensory foraging variables, especially when one of the species is a folivore and is fully trichromatic.

We thank the reviewer for highlighting this point. We realize that we needed to present with greater clarity the study design with respect to the interspecies comparisons. As the reviewer correctly points out, species/ diet effects could be conflated with color vision variation. To avoid this, for Question 1 “**Do primate species vary in their fruit investigative behaviours reflecting dietary specialization or sensory anatomy?**”, we only included data from *trichromatic individuals* when examining the effects of primate species/ dietary specialization. I.e., for Question 1 analyses, we included records from all howler monkeys, and only records from the spider monkey and capuchin monkey individuals who had trichromatic color vision.

This approach was not clearly presented in the initial version. We have now clarified our text.

Lines 234-235: “To address **Question 1**, we only included data from trichromat individuals (n = 986 feeding bouts, 14,328 investigation sequences, Table S3) to avoid confounding influence of polymorphic colour vision on interspecific variation.”

1.3. Since it appears that individual color vision genotypes are known, are there differences in the sensory investigation variables among dichromatic vs trichromatic females within the polymorphic species? Do these parallel the findings of the study as a whole (e.g., more sniffing in the dichromats)? Even if the sample sizes are low for such a comparison, the specificity of the contrast

would provide additional confirmation of the larger findings, while alleviating some concerns about potential differences in sex, site, and observation period.

We have now added, as supplemental material, analyses comparing dichromatic females to trichromatic females within the two polymorphic species (*Cebus*, *Ateles*), as requested. Overall, the results of this analysis are very similar to those observed when using the full dataset (new Table 1, Table S5, Fig. S3). We identified main effects of colour vision phenotype and an interaction effect for sniffing behaviour, with dichromats using sniffing more often than trichromats, particularly when feeding on cryptic fruits. Similarly, we identified an interaction effect for bite/reject during fruit investigation. In the full dataset, we also found a main effect of colour vision phenotype on bite/reject. In this more limited dataset the main effect only approached significance ($p = 0.064$).

As in the full analysis, we also identified significant interactions between colour vision phenotype and fruit conspicuity for the use of manual touch and fruit acceptance in this dataset of polymorphic females (Table 2, Table S5, Fig. S3). We also identified new significant main effects of fruit conspicuity. Specifically, in this female-only dataset for *Ateles* and *Cebus*, we also found that manual touch was used significantly more frequently when foraging on cryptic fruits, and that cryptic fruits were less likely to be accepted relative to conspicuous fruits. We have added a description of these new results to Supplementary Materials (Supplementary Results S3, Table S5, a new Fig. S3) and describe them in the main text.

Lines 318-324: “To address confounding influences of sex and routine vs. polymorphic trichromacy on these analyses, we repeated this analysis with only females of polymorphic species (*Ateles*, *Cebus*) and identified the same pattern (Supplementary Results S3, Table S6, Fig. S3), but also identified significant main effects of fruit conspicuity on the use of manual touch and fruit acceptance. Among *Ateles* and *Cebus* females, cryptic fruits were touched more often and were accepted less often relative to conspicuous fruits.”

1.4. Furthermore, are there any differences among the variables of interest between the two *Alouatta* sites?

We have now added a new analysis comparing foraging sensory behaviours by howler monkeys for the fruit species that were eaten at both sites (Supplementary Results S1). In addition, we collected additional behavioural data in Santa Rosa in May- August 2021, allowing the datasets to be more robust and better facilitate this comparison. We find that howler monkeys at the two sites differ significantly in their use of manual touch and bite/reject during foraging (Mexican howlers use both behaviours more frequently), but howlers at different sites do not differ in the use of sniffing. However, the pattern and significance of differences between the three primate species remains the same when analyses are repeated with just the Sector Santa Rosa *Alouatta*, just the Isla Agaltepec *Alouatta*, or the combined howler dataset. We conclude that the intraspecific variation in *Alouatta* due to site differences in ecology or behavioural traditions is much smaller than the interspecific variation we observe.

We have added these results and discussion to Supplementary Results S1 and describe them briefly in the main text (lines 300-303): “We detected variation in the use of touch and bite/reject between *Alouatta* from the different sites, this variation did not influence the pattern of interspecific differences (Supplementary Results S1), and there was no effect of sex on the use of sensory behaviour in *Alouatta* (Supplementary Results S2).”

1.5. Given the above, it may make for a “cleaner” signal if *Alouatta* were excluded from the initial analysis.

We appreciate the motivation for this suggestion. However, the comparison of howlers to other primates is central to research Question 1. As an alternative to excluding howler monkeys, we examine only trichromatic individuals of all species when evaluating species differences. We now more clearly explain and justify this approach (see reply above), and add additional analyses to make sure that sex biases and site differences do not influence the howler monkey values in the comparative analysis.

1.6. Specific comments by page [Note: There is a discrepancy between page numbers at the top and bottom right of the page. I am using the page numbers at the top throughout this review.]

We apologize for the confusion introduced by the conflicting page numbers and have made sure to remove our own page numbers prior to resubmission so that this doesn't occur again.

1.7. Abstract, line 57: should read “...that THE species with the highest...”

Corrected, thank you!

1.8. p. 6, lines 103-106: While I appreciate that there may be limits on manuscript page numbers, there are some omissions of primary literature, such as Jacobs et al. (1996) as a reference for routine trichromacy in *Alouatta*.

Jacobs GH, Neitz M, Deegan JF II, Neitz J (1996) Trichromatic colour vision in New World monkeys. *Nature* 382:156–158.

We have corrected this omission and have added this reference. Thank you.

1.9. p. 8, paragraph starting on line 156: Please provide more details on the focal follows: how often were the groups of each species followed? Were these all day follows? Were females and males followed approximately equally?

We thank the reviewer for this suggestion. We have now expanded the details on our observation schedule. Due to space constraints, some of these details are now presented in Supplemental Methods. In addition, we now also highlight Supplemental Table S1 where motivated readers can carefully examine the breakdown of our data by any variables of interest (primate species, age, sex, colour vision phenotype, study site).

In support of these revisions, we have added the following details to the text:

Lines 142 (Main Text): “Observation details are presented in Supplementary Methods S1 and Supplementary Table S1.”

Lines 145-147 (Main Text): “Individuals were sampled opportunistically, based on visibility, but we rotated among sex and age classes in an effort to sample evenly across these variables.”

Supple Methods S1: Data collection typically occurred during full-day follows, although occasionally we followed groups for partial days if extreme weather, terrain, or other factors caused us to lose track of the primate social groups. Social groups of each species were followed 12-20 times per month of data collection.

1.10. p. 8, (b) Dietary specialization: Do the authors' own data agree with the literature feeding data? There is quite a range in the intraspecific frugivory percentages among the studies cited. The current comparisons are limited to foods that were foraged in common by the three species at both sites (p. 11); however, the large range of values associated with each species in terms of monthly frugivory suggests that there are interannual differences in fruit productivity even within a site that could affect results. As described on pp. 7-8, the study periods in Santa Rosa were largely non-overlapping for the three species, and *Alouatta* was studied at two different sites.

Our data collection was based on short, targeted focal animal follows occurring within fruit patches. We sought to document specific foraging sequences, investigative behaviours, and feeding rates and we did not consistently nor comprehensively collect data on activity budgets or all-occurrence visits to fruit patches by all primate species in this study. As such, the data collected for the present study are not appropriate for answering this question. (Although, our personal experiences confirm this pattern, and the relative rarity of fruit tree visits by *Alouatta* is one reason we have less data for this species).

Fortunately, Chapman (1987, reference 53) conducted a study specifically exploring the diets of *Ateles geoffroyi*, *Alouatta palliata*, and *Cebus imitator* at this study site in Santa Rosa. When we examine the 12 months where dietary data were consistently and concurrently collected by Chapman - and fruit productivity was therefore experienced similarly - for all three monkey species (January-July 1984 and February-June 1985), the average monthly frugivory was ~74% for *Ateles*, ~67% for *Cebus*, and ~29% for *Alouatta*. This is in agreement with the broader literature examining frugivory by these three primate genera. [Please note that the exact monthly frugivory values were not provided in Chapman (1987), but we extrapolated these data from its Figure 2 (percentages of observed feeding time that each species spent eating fruits, flowers, leaves, and insects during each month of the study) using ImageJ.]

We have now added the following sentence to our manuscript, in case other readers have the same question:

Line 272-274: "These results are also consistent with feeding records of these three sympatric species at SSR when they are observed concurrently during the same months and years [53]."

1.11. p. 9, Olfaction paragraph: The authors mention that absolute MOB volume is an indicator of olfaction. However, nasal turbinate surface area probably scales with respect to body mass. Did the authors take body mass differences into account when comparing nasal turbinate surface area across species? [Not as relevant to take body mass into account for thumb:index finger measurements since this is a ratio.]

We thank the reviewer for highlighting this concern. We have now rerun the nasal turbinate surface area analysis using a size correction of skull geometric mean. We calculated a relative measure (surface area/skull geometric mean), and reran the Kruskal-Wallis and Dunn tests. Our results were the same, and we have added a statement to the results section.

Line 282-283 “This result persists even after controlling for differences in skull size (KW: $\chi^2 = 16.54$, $p < 0.0001$).”

We have applied the same correction for fungiform papillae density and added these results.

Lines 290-291: “ If body size is controlled for, the pattern remains, except *Cebus* also has higher papillae densities than *Ateles* ($p = 0.026$).”

1.12. p. 10, Colour vision type paragraph, first sentence: By “well established,” do the authors mean that the color vision genotypes of each individual in their *Cebus* and *Ateles* groups are known?

Yes - this is what we meant. We have now revised the sentence for clarity:

Line 208-209: “Colour vision genotypes for all individuals in our SSR study population have been previously reported [5,44–46]”.

1.13. p. 11, line 241: missing word—“...datasets to include only those plant species in WHICH all three primates species foraged...”

We thank the reviewer for highlighting this. We have revised that sentence following advice from Reviewer 2, so it now reads:

Line 224-225: “We restricted behavioural analyses to only include foraging sequences on plant species consumed by all three primate species.”

1.14. p. 11, lines 251-252: This is a bit confusing. It is not clear what data “...included data from trichromat individuals” is in reference to. It seems to mean (based on later descriptions of models on the next page) data on the total numbers of fruits that were foraged by trichromatic individuals but not that only trichromatic individuals were being compared across the study species? If the latter, then how could dichromats vs trichromats be compared in the GLMM?

We have revised this section for clarity and to specifically explain the datasets used to address Question 1 (Do primate species vary in their fruit investigative behaviours) and Question 2 (Does color vision phenotype influence the use of other sensory behaviours). Specifically:

- Lines 234-236: “To address Question 1, we only included data from trichromat individuals ($n = 986$ feeding bouts, 14,328 investigation sequences, Table S3) to avoid confounding influence of polymorphic colour vision on interspecific variation.”
- Lines 250-251: “For Question 2, we included data from both trichromat and dichromat individuals of *Cebus* and *Ateles*, and all *Alouatta* individuals.”

We hope these clarifications now eliminate the confusion.

1.15. p. 11, lines 253-261: Only the results of a likelihood ratio test between a full model and a null are provided in the supplemental materials, but not a summary of the full model. What are the results for the fixed effects of the full model and the pairwise interactions of their levels (it is fairly standard to report pairwise contrasts with the emmeans package in R)? What are the effects of the random variables?

We have now added a supplementary table (Table S5) which includes the parameters (estimates, standard errors, Z-values, and p -values of fixed effects and variance of random effects) for the GLMMs of the behavioural data for Question 1 (interspecific variation in sensory behaviours). We continue to also report the results of the likelihood ratio tests and the pairwise contrasts in Figure 2. We have also added a new table (now Table 1) that reports model estimates, LRTs, and pairwise contrasts for Question 2 (colour vision effects), as well as a supplementary table (Table S6) that reports the same data for analyses including only *Ateles* & *Cebus* females.

1.16. p. 12, top of page: What is the effort expended (e.g., bouts, time spent) by each of the three species on conspicuous vs cryptic fruits?

Unfortunately, we are not able to comprehensively answer this question. As we explain above regarding the question about dietary specialization (see Comment #1.10), we can't answer questions about effort/ activity budget from our sampling scheme. Our data collection was based on short, targeted focal animal follows occurring within fruit patches. We sought to document specific foraging sequences, investigative behaviours, and feeding rates and we did not consistently nor comprehensively collect data on activity budgets or all visits to fruit patches by all primate species in this study. However, motivated readers are able to examine our full dataset on Dryad and from this see the distribution of data across different fruit types. This likely is a rough proxy for effort expended, but because the volume of data recorded in each fruit species may be affected by visibility or other factors (for example, less data collected from fruit trees where visibility was poorer, such that relatively longer observational time was required to get the same amount of data), we prefer to not use this dataset to answer this question in the manuscript.

1.17. p. 13, paragraph on foraging behaviour: references for figures should be for Fig. 3 not 2. This figure is difficult to interpret quickly. It would help if the photos (or just labels) of the monkeys were positioned next to their actual positions on the plot.

We appreciate this feedback. We have revised this figure and moved the labels and photos to hopefully allow quicker interpretation.

1.18. This is not related to a specific result, but I ask here since it fits best with questions on foraging: are there sex differences within species with respect to the sensory foraging variables? This would be interesting to show because only females are fully trichromatic within the species that are polymorphic for color vision. Furthermore, are there differences between fully trichromatic and dichromatic females?

We have performed a new analysis to investigate sex differences in sensory behaviours across the full dataset (as well as only in *Alouatta*, Supplementary Results S2, also see Comment #1.5 above). After controlling for primate species and colour vision phenotype, we did not identify any significant effects of sex on the use of sniffing, manual touch, bite/reject, or fruit acceptance. These new analyses are now reported in Supplementary Results S4.

As we explained above in response to Comment #1.3, we have also added an analysis to Question 2, where we limited the investigation of colour vision effects on sensory behaviours to just dichromat and trichromat females of polymorphic species (*Ateles*, *Cebus*: Supplementary Results S3). Our results are

broadly consistent between the full dataset and the subset including only females of polymorphic species, and discussed in detail in response to Comment #1.3, and Supplementary Results S3.

In the spirit of being very comprehensive, we have also added a new test to look for differences between male and female howler monkeys (which are both routinely trichromatic). We find no difference between male and female howler monkeys in sensory behaviours. This additional test, with an accompanying explanation, is now included in lines 302-303 "... there was no effect of sex on the use of sensory behaviour in *Alouatta* (Supplementary Results S2)".

1.19. p. 13, line 318: Figs. S5 and S6 are misnumbered—there is no Fig. S4.

We appreciate being alerted to this. We have corrected the numbering of the supplementary figures.

1.20. p. 13, lines 321-323: "Dichromats bit and rejected fruits....2.7 times the rate of trichromats for cryptic fruit..." From Fig. S5, this appears to be true only for *Ateles* but not *Cebus*.

The incident rate ratios reported were those calculated by the GLMMs, which controlled for differences in each sensory behaviour associated with plant species/morphotype and individual identity, while the data in Figure S5 represent raw data. We apologize for the confusion, and we have removed Figure S5 from the Supplementary Material because we feel it creates more confusion than benefit. Instead, we have added a new supplementary figure (Fig. S3) which reports the marginal effects of the interactions of colour vision phenotype and fruit conspicuity from the models for each sensory behaviour. We have also revised the wording of this section and removed the incidence rate ratios.

1.21. p. 16, line 402: should read, "...use of other senses IS intriguing."

This is not a particularly unexpected finding given that routine trichromatic color vision is negatively associated with olfactory receptor genes (Gilad et al., 2004).

Gilad Y, Wiebe V, Przeworski M, Lancet D, Pääbo S. 2004. Loss of olfactory receptor genes coincides with the acquisition of full trichromatic vision in Primates. *PLoS Biology* 2 (1): 120-125.

We thank the reviewer for flagging this - we have corrected this typo. We also appreciate the suggestion regarding the link to OR genes. However, this story is proving more complicated than presented in this 2004 paper. Two authors (Wiebe V and Pääbo S) retracted their names from this publication in 2007 (*PLoS Biol* 5(6): e148. doi:10.1371/journal.pbio.0050148), and other papers have found no clear relationship in anthropoid primates (Matsui et al. 2010 *Molecular Biology and Evolution* 27: 1192-1200; Niimura et al. 2018 *Molecular Biology and Evolution* 35: 1437–1450). Discussion of the complicated olfactory genotype/ phenotype is beyond the scope of our study and so we prefer not to bring it in here, as we don't have the space to do it justice, but we hope the present paper will motivate future efforts to look at this.

Referee: 2

Comments to the Author(s)

In this study, Melin and collaborators examine the relationship between foraging behavior and sensory anatomy in three species of neotropical monkeys. The study is highly integrative, combining detailed field observations with advanced tools to examine anatomy (e.g., micro-CT scanning). The authors

show differences in use of olfaction and touch among species that differ in their degree of frugivory and, interestingly, between dichromatic and trichromatic individuals. Overall, the study is an exciting contribution to the fields of behavioral ecology, functional morphology, and primatology.

The manuscript is well written and straightforward; the study methods and analyses as well as the interpretations of the results seem appropriate. My detailed suggestions and questions are shown below.

2.1. L56: the abstract needs an overarching hypothesis.

We agree and have added the following in lines 55-56: “We hypothesize that dietary and sensory specialization shapes foraging behaviours as an adaptation to assess food quality”

2.2. L71 (and 52-53): this statement needs some refinement about the gaps; e.g., echolocation in bats and cetaceans, and lateral-line sensing have been studied extensively.

We appreciate this suggestion and agree.

We have revised the line in the abstract (lines 51-52) to specify: “research on the senses used during foraging by wild terrestrial vertebrates is sparse.”

We have also revised the introduction to state:

Lines 68-70: “A large body of work has explored the role of colour vision in finding and assessing foods in terrestrial vertebrate taxa [3–5] but the role of non-visual senses during foraging has received considerably less attention, with the notable exception of chiropterans [6].”

2.3. L73: animals terrestrial frugivores

We have replaced “animals” with “terrestrial frugivores”. (Line 52)

2.4. L97: for those not familiar with primates, it would be useful to add a couple of sentences about the major primate clades (catarrhines, strepsirrhines, platyrrhines) and their vision types. This could even extend to mammals in general, to put into broader context.

We have edited this paragraph to better contextualize primate colour vision, as the reviewer suggested. (Lines 91-105)

2.5. L162: I appreciate the inclusion of these videos in the supplement, they are very useful!

We are very pleased to hear this!

2.6. L171: clarify if these data were restricted to the study localities, or whole species ranges, and what other parameters (e.g., search engines, terms, years) were used.

We now clarify that these data were collected from studies conducted throughout the ranges of the species of interest, and include the search engine and terms used. We also clarify how we handled the date ranges.

Lines 158-163: “We searched for studies on Google Scholar using the terms “Ateles geoffroyi”, “Cebus capucinus”, “Cebus imitator”, “Alouatta palliata”, “diet”, and “frugivory” without applying date restrictions. We included studies conducted throughout the geographic ranges of each species that used focal animal sampling to collect feeding data from study groups inhabiting relatively undisturbed forest, for a period of at least six months.”

2.7. L192: clarify which turbinate bones (I see this is in the supplement, but it would be useful to present in the main text); the reference provided is for marmosets – is this extrapolation realistic? Add a caveat if necessary.

We have also added information to the methods on which turbinate bones were measured. Smith and colleagues have published extensively on functional groupings of turbinates. While they have not sampled as broadly in platyrrhines as they have in catarrhines, their work suggests that this extrapolation is a reasonable one to make. However, we agree the reference was narrow. In response to this comment, we have changed the reference to Smith et al. (2004), which is less specific to marmosets and applied more broadly across primates:

Smith TD, Bhatnagar KP, Tuladhar P, Burrows AM. 2004 Distribution of olfactory epithelium in the primate nasal cavity: are microsmia and macrosmia valid morphological concepts? *Anat. Rec. A. Discov. Mol. Cell. Evol. Biol.* **281**, 1173–1181.

Lines 178-185: “We measured the posterior/superior row of turbinates that are covered in olfactory epithelium *in vivo* (ethmoturbinate, nasoturbinate_[39]) using computed tomography (CT) scans [40]. We downloaded CT scans ($n = 8$ *Alouatta palliata*, 15 *Ateles geoffroyi*, 2 *Cebus imitator* (previously *C. capucinus*: [38])) from the digital repository Morphosource.org (Supplementary Dataset 2). Using Avizo 9.1 ® (<http://www.fei.com/software/avizo3d/>), we digitally extracted olfactory turbinates from each specimen, visualized turbinates as 3D surfaces, and measured the surface area volumes (Supplementary Fig. 1). We also collected data on skull size (calculated as the geometric mean of cranial length and width) for each specimen.”

2.8. L197: should “volumes” here be “areas”? If not, this does not seem appropriate since the area is a better proxy for the olfactory epithelium. Did these analyses also include a size correction? If not, I recommend re-running additional analyses correcting for size (e.g., nasal cavity volume or even body mass) and reporting in the supplement.

We appreciate this comment and apologize for the lack of precision in our terminology. The correct term for our analysis should be “area” and we have revised this throughout.

In response to the second point, we have now run an analysis using a size correction of skull geometric mean. We calculated a “relative nasal turbinate surface area” as the raw surface area divided by the skull geometric mean, and reran the Dunn test. The results were the same, and we have added a statement to the results section (see Comment 1.11).

2.9. L201: justify why 0.5 cm of the tongue was measured.

For these data, we are reporting published fungiform papillae density counts for these three species from Alport (2009) and not our own measurements. We did not choose the 0.5 cm location, but believe Alport chose the anterior 0.5 cm of the tongue because it was accessible for measuring in live animals

as well as cadaveric ones, thus permitting a larger comparative sample, and the anterior tongue in general exhibits the highest fungiform papillae density. In general, research on fungiform papillae density in humans is also calculated from the anterior tongue [e.g., reference 33].

We have revised the text to make this clearer (Line 187-191): “Fungiform papillae in the tongue house taste receptors, and increased density of these papillae on the anterior tongue is positively associated with taste sensitivity [33]. We obtained published data [40] on the density of fungiform papillae on the anterior 0.5 cm of the tongue and body mass for 14 howler (*Alouatta palliata*), 4 spider (*Ateles geoffroyi*), and 12 capuchin (*Sapajus [Cebus] apella*) monkeys ...”

However, we appreciate the opportunity to think about the impacts of the size of the area sampled. We have added an analysis to control for effects of body size. The results are comparable, although the difference between *Cebus* and *Ateles* is now significant as well. The results of this test are now in the Main Text (see Comment 1.11).

2.10. L204: add which species.

We have combined the two sentences relevant to this comment to now read (lines 188-191): “We obtained published data [40] on the density of fungiform papillae on the anterior 0.5 cm of the tongue and body mass for 14 howler (*Alouatta palliata*), 4 spider (*Ateles geoffroyi*), and 12 capuchin (*Sapajus [Cebus] apella*) monkeys (because data were not available for *Cebus imitator*)”.

2.11. L226: “species” here should be “individuals”, correct?

Yes, thank you. We have revised accordingly.

2.12. L240-241: sentence needs editing for clarity.

We have revised this sentence for clarity as follows:

Line 224-225: “We restricted behavioural analyses to only include foraging sequences on plant species consumed by all three primate species”

2.13. L259: were these analyses run with different reference species? Do the results change? If so, this needs to be acknowledged and discussed.

We thank the reviewer for this point. We did rerun the analyses with different reference species and the results do not change. We have added a sentence to that effect in the Methods (line 247): “The choice of the reference species did not change the results.”

2.14. L333: these results are promising given the coarse morphological proxies used for olfaction (given the complexity of this system). The authors could acknowledge other additional factors that could potentially blur the relationship between olfactory anatomy and behavior, including that (1) the shape of the turbinates and nasal cavity affects air flow inside the nose and odorant deposition, (2) the density of olfactory neurons in the OE and expressed ORs may vary. Additionally, could the

authors discuss if /how these fruits change in aroma when they are ripe, and how that might relate to behavioral differences? If unknown, this could be mentioned as a future direction.

We agree and appreciate this suggestion. We have added the following sentences to the end of the olfaction section. (We unfortunately needed to keep this rather brief, to accommodate the other additions requested, given the strict limits on space.)

Line 345-349: “Many factors influence a species’ olfactory ecology, including the shape of the turbinates and nasal cavity, olfactory neuron density, olfactory receptor diversity, and their responsiveness to natural odours [25,30,55]. Future studies integrating these variables, and increasing the taxonomic diversity studied, will further illuminate the relationship between olfactory anatomy and behavior.

Appendix B

Review of, “Anatomy and dietary specialization influence sensory behavior...,” by Melin et al. for Proceedings B.

This manuscript attempts to do something very rarely done in the literature, which is to combine behavioral data with anatomical correlates in an attempt to explain the links between the two. For this, the manuscript is to be admired. Furthermore, the datasets involved are strong, especially the behavioral foraging sequences. The authors also do an admirable job of attempting to make sense of these diverse and copious data.

However, there are certain conceptual and practical issues that I have with the manuscript in its current form. I detail these below, and I hope the authors find them useful.

Major Comments

1. Some of the correlates of the olfactory sense are problematic. First, the use of absolute MOB size has some issues, and the justification for its use is a single study on individual variations within humans only. This study cannot be extrapolated across species. Indeed, the olfactory bulb should be at least in part a function of the size of the animal (or of the nose more specifically), as a larger nose is filled with more olfactory mucosa, which thereby result in an increased OB size simply because that’s where the axons of the receptors all terminate. Second, the authors should have used a histological examination of OE surface areas, because simply comparing the sizes of the superior/posterior turbinates are not enough, as it is known the different species have different percentages of a given turbinate covered in OE.
2. Regarding the correlates of taste (lines 189-190), is the anterior 0.5 cm of the tongue the same *relative* amount of the tongue across the 3 species? It would seem like this should be standardized, and I’m sorry if I missed that it was.
3. The findings in Question 2 are interesting, but I was hoping to see purely a (statistical) comparison between di- and tri-chromatic *Ateles* and *Cebus* only—i.e. without *Alouatta*, because it seems like the data from the howler monkeys could influence the results in a disjointed way, since they are only represented as tri-chromatic animals. Did the authors do this and I’ve missed it?
4. Unfortunately, Discussion point “c” is too heavy-handed based on the results of this paper. The paper has various strengths, but taxonomic breadth is not one of them. Thus, I don’t think the results from this study are enough to call into question, e.g., proxies used for olfaction. For example, larger MOB volume is *not* associated with active sniffing (Line 411), and I don’t even know what such a statement would mean. Further, larger turbinate SA has been associated with many aspects of olfaction, and the support for it being related to sensing faint odorants is very scant and not conclusive (again, human studies are not really appropriate in this context). Larger turbinate SAs could also be related to the breadth of odors that can be sampled, as that surface gives more space for a sufficient quantity of diverse OR receptors to be expressed. Whatever the case, we’re still on very tentative ground here, and this study doesn’t really change that.

Other Specific Comments

Lines 67-74: There’s a substantial literature on fruits and olfaction in rodents, no?

Line 149: Was sniffing simply coded as binary (i.e. occurred or not), or was the duration of sniffing taken into account?

Discussion: It would be nice for the authors to have a preliminary paragraph that summarizes the major points, and then they go into detail later. I don't really think that breaking the discussion down into Question 1 and Question 2 is necessary.

Lines 336-338: What does this sentence actually mean? i.e. the MOB *always* processes odors, during "active sniffing" or not, so I'm really not sure what the authors are trying to say here.

Lines 341-342: I think the authors are misinterpreting this paper. Passive smelling of course activates brain regions that differ from active sniffing-induced smelling. But, both types of smelling always activate the MOB. So, I'm not sure what the authors are arguing for here, either.

Throughout: Please make sure to use italics for genus/species names throughout. Also, in my opinion it's better to just use the Scientific names rather than going back and forth between scientific names and common names, as much of the readership may not be familiar with primates at all.

Figure 2: I find these plots very confusing. Are there other, more intuitive ways of plotting these data?

Appendix C

08-Mar-2022

We have now received referees' reports on your revised manuscript RSPB-2022-0048 entitled "Anatomy and dietary specialization influence sensory behaviour among sympatric frugivorous primates". This was reviewed by one of the original referees and also by a new referee.

We are all agreed that this is a very interesting topic, and that the revised manuscript is much improved. However both referees of this version still have substantial concerns which will require major revision. I am therefore rejecting the manuscript in its current form, but inviting a resubmission, provided the comments of the referees are fully addressed. We do not usually allow a second round of major revisions, but will be willing to consider this if you can address all points. Please note in particular the need to acknowledge the limitations of the study in terms of the general conclusions that can be drawn. I must also emphasise that this is not a provisional acceptance.

We are pleased our manuscript was found to be much improved and we are grateful for the opportunity to resubmit our manuscript. It is unfortunate that only one of the reviewers agreed to examine the revised version, but we appreciate the feedback from a new, third expert, which has allowed us to strengthen our manuscript in new directions.

We have revised our manuscript to better acknowledge the limitations of the study and avoid drawing premature conclusions. We have re-written the olfaction section of our discussion (lines 336-359) and the final paragraph of the manuscript (lines 415-428).

We have also made other revisions in line with the suggestions made by the editor and reviewers, which we detail below.

- 1) A 'response to referees' document including details of how you have responded to the comments, and the adjustments you have made.
- 2) A clean copy of the manuscript and one with 'tracked changes' indicating your 'response to referees' comments document.
- 3) Line numbers in your main document.
- 4) Please read our data sharing policies to ensure that you meet our requirements <https://royalsociety.org/journals/authors/author-guidelines/#data>.

To upload a resubmitted manuscript, log into <http://mc.manuscriptcentral.com/prsb> and enter your

Author Centre, where you will find your manuscript title listed under "Manuscripts with Decisions." Under "Actions," click on "Create a Resubmission." Please be sure to indicate in your cover letter that it is a resubmission, and supply the previous reference number.

Yours sincerely,
Professor Loeske Kruuk
Editor
mailto:proceedingsb@royalsociety.org

Associate Editor Board Member

Comments to Author:

Thank you for resubmitting your manuscript to Proceedings B. We have now received two reviews of your manuscript, one of which is from a reviewer of your initial submission. Both reviewers agree that this study addresses a question that is important and rarely investigated in wild primates. They also greatly appreciate the enormous effort that went into obtaining these data. Unfortunately, the reviewers still raise issues that will require substantial revisions to address. Both reviewers find that some of the conclusions in the manuscript are stated more strongly than can be supported by the results. One of the reviewers had concerns about the interpretation of the results of some cited literature, specifically how active sniffing and passive smelling are processed in the brain. This reviewer was also concerned about the justification for using certain measurements, e.g. absolute vs. relative MOB size and turbinate surface area without knowing the proportion of that area covered in olfactory epithelium. The mix of supported and unsupported predictions could be related to the choices for measurements included in the analyses. I hope the feedback from the reviewers is helpful to you for a future version of the manuscript.

We are grateful for the supportive words about the importance of our research and the acknowledgement about the difficulty of collecting all of these diverse data types! We have revised our manuscript to address the few outstanding concerns raised by reviewer 1, and to address for the first time the suggestions of reviewer 3. We feel our manuscript is much improved, and hope it is now found acceptable for publication.

Reviewer(s)' Comments to Author:

Referee: 1

Comments to the Author(s).

General comments

Melin et al. investigate sensory foraging within and across three platyrrhine species that differ in dietary specialization and colour vision phenotype. The authors pose two main questions: 1) are there differences in foraging behaviours (sniffing, touching, biting) among their study species related to degree of frugivory and sensory anatomy (olfactory bulb volumes and turbinate surface areas, fungiform papillae density, and thumb/finger ratios)?, and 2) is colour vision phenotype related to the foraging behaviours of interest and the accuracy of food selection? These questions are compared using two datasets: trichromatic individuals of all three species for the first question and all individuals for the second. In this revision, the authors have performed additional analyses (in the supplementary material) that compared the howler monkeys from the two different sites

(for Question 1) and compared the females in the species polymorphic for colour vision for Question 2.

The authors' primary findings are that the most frugivorous species (*Ateles*) sniffs fruit more and the most manually dexterous species (*Cebus*) touches fruit the most. Main olfactory bulb volume is a better predictor for fruit sniffing than nasal turbinate surface area, which leads to an interesting suggestion that these two olfactory morphologies are responsive to different stimuli. With respect to colour vision phenotype, the results show that across all three species and for females within the polymorphic species, dichromats sniff and bite fruits more than trichromats, and trichromats touch and accept cryptic fruits more than dichromats.

The manuscript is well-written and detailed and clearly lays out the many datasets that were used in the analyses. I have to applaud the tremendous effort by the authors to add supplementary results to reinforce their initial conclusions. In particular, the authors have added a supplemental analysis comparing dichromatic and trichromatic females within the polymorphic species that show that the signal is similar to (or in the same direction as) the findings of the larger dataset. The new analysis (Fig. S3, Table S6), in which the authors excluded *Alouatta*, have addressed my prior concerns and have reinforced the results of the larger dataset on how colour vision phenotype interacts with the behavioural variables measured. An added benefit of analysing the polymorphic species is that the comparisons are confined to the frugivorous taxa. The numbers of different types of data (foraging behavioural traits, sensory anatomy, colour vision phenotypes) compared and integrated here will make this a valuable contribution to the field and a good jumping off point for further research.

We thank the reviewer for their supportive comments and we are pleased that we have satisfactorily addressed the concerns raised in the first round of revision.

Specific comments by line

Title: "frugivorous primates" suggests that the taxa studied are all frugivores. I don't think *Alouatta* would be considered one...

We take the reviewer's point and have revised our title to remove the word frugivorous. "Anatomy and dietary specialization influence sensory behaviour among sympatric ~~frugivorous~~ primates"

Abstract, line 54: "We find the most frugivorous species sniffs fruits the most often..." While this is not wrong, it is a bit misleading since most of these foraging behaviour variables are not well associated with degree of frugivory. In terms of sniffing, the pattern is related to the relative size of the MOB in the three species. Perhaps this could be rephrased?

We have now revised the sentence to soften the statement. We replace "the most frugivorous species (*Ateles geoffroyi*)" with "frugivorous spider monkeys (*Ateles geoffroyi*)". We hope this is acceptable as the latter is a standard way to describe this group of primates, and we prefer to keep this reference to the diet in the sentence in order to better link our results to the structure of the research questions (i.e. evaluating the impact of diet). To match the new structure, we add "omnivorous capuchins" to the abstract when discussing the result about *Cebus* later in the sentence. We recognize that diet likely isn't the full story in our manuscript. We have made extensive changes to the discussion (re-written the olfaction section (lines 336-359) and the final paragraph (lines 415-428)) to better caveat the limitations of our study.

Line 138: reference to Fig. 1 for study species is no longer applicable.

We appreciate the reviewer catching this. We detected that we actually do have space to retain Figure 1 as a small figure, and have re-added it to our manuscript. We feel it helps non-specialist readers become more familiar with the study species as a reviewer mentioned the audience might not be familiar with these primates.

Line 232: I asked previously about the effort expended (e.g., bouts, time spent) by each of the three species on conspicuous vs cryptic fruits. I agree that the overall percentages of conspicuous vs cryptic fruits in the diet are not germane to the question of fruit ripening. However, understanding the relative contributions to the diet of these fruits would affect how one would interpret the importance of associated behaviours. For example, even if the monkeys used manual touch more frequently on cryptic foods and rejected them more often (at least among the dichromats), if cryptic foods are not eaten that frequently to begin with, is this really that relevant? Could the authors provide sample sizes in the ms. for conspicuous and cryptic fruits used in their analysis to get at a sense of the relative contributions of these foods to the diet? (I realise the full dataset is in the Dryad repository.)

We are happy to do this and have now added a supplementary table (Table S7) that lists sample feeding bout and foraging sequence sample sizes and percentages for conspicuous and cryptic fruits used in our analysis. We refer to this table directly to the manuscript:

Lines 312-314:

“Across all three species and both color vision phenotypes, conspicuous fruit were the most common focus of feeding bouts (70-98% of bouts, Supplementary Table S7).”

Line 234: missing word: “...we only included data from trichromatic individuals...to avoid THE confounding influence...”

We thank the reviewer for noting this and have added “the” to our revised manuscript. (line 239)

Line 267: dietary specialization results. I asked in the prior version of this ms. about the literature feeding data. In the Supplementary Datasets that the authors nicely provided, frugivory percentages for the same month vary widely between studies of the Cebus monkeys (this does not appear to be such an issue with Alouatta or Ateles). Given this high variance, it may not be that surprising that the behavioural variables examined were not well-associated with degree of frugivory.

Capuchin monkeys indeed have a higher dietary flexibility, which has been attributed as a flexible responses to seasonal shifts in food availability (reviewed in Melin et al 2020). We are now more cautious about our interpretations, mention Cebus dietary variation as a possible source of noise in the data, and call for additional research.

Lines 339-342. Accordingly, our results supported our prediction based on relative MOB anatomy, but not necessarily dietary specialization. However, given the limitations of our sample size, and the large dietary flexibility present in *Cebus* [37], which may complicate diet-sensation relationships, more extensive study of a greater number of species should clarify the relationships between diet and olfactory specialization.

Line 268: Though the results of the statistical tests for all the results shown in Fig. 1 are given in the text, could these be included in the figure caption or added to Supplementary Table S4?

We appreciate this suggestion and have now added these to the figure caption as requested.

Line 300: I asked previously if there were any differences among the variables of interest between the two *Alouatta* sites? The authors have added a new analysis to address this (Supplementary results S1). The howler monkeys at the two sites differed considerably in touch and biting/fruit reject. When compared to the other taxa in the larger dataset, either individually or combined, however, the authors found no differences in the results. Again, this gives greater confidence in the overall interpretation of the larger dataset, and I appreciate the additional effort made for this comparison.

We are very pleased to hear this! Thank you.

Line 366: this is supposed to be supplementary Fig S4? This figure is not strictly necessary, but if it is included, then a figure of either a capuchin or howler monkey should also be added as a comparison.

This sentence was supposed to reference Fig S4 - our apologies for the confusion. We have now added lateral and frontal pictures of a capuchin monkey for comparison, and updated the reference to this figure. Unfortunately, we don't have comparable pictures of a howler monkey.

Figures:

Fig. 1: Please add results of statistical tests to caption.

We have added the statistical test results to the caption and removed them from the figure itself.

Fig. 2: Thank you for repositioning the species photos next to their plot positions—it makes it easier to read. Perhaps add to the caption that the dataset is for Question 1 with trichromatic individuals of all three species in the analysis.

We agree and have made this revision.

Fig. S3: Authors changed plot types from incident rate ratios to marginal effects plots of interactions. This new figure is much easier to interpret.

We are pleased to hear this.

Fig. S4: not referenced in manuscript, but also not strictly necessary.

We have updated the reference to this figure, which was incorrectly labeled at Fig S6 by accident in the previous version.

Other:

Results S4: not referenced in manuscript, but also not strictly necessary.

We have added a reference to this in the manuscript (Line 376)

Referee: 3

Comments to the Author(s).

Review of, “Anatomy and dietary specialization influence sensory behavior...,” by Melin et al. for Proceedings B.

This manuscript attempts to do something very rarely done in the literature, which is to combine behavioral data with anatomical correlates in an attempt to explain the links between the two. For this, the manuscript is to be admired. Furthermore, the datasets involved are strong, especially the behavioral foraging sequences. The authors also do an admirable job of attempting to make sense of these diverse and copious data.

However, there are certain conceptual and practical issues that I have with the manuscript in its current form. I detail these below, and I hope the authors find them useful.

Major Comments

1. Some of the correlates of the olfactory sense are problematic. First, the use of absolute MOB size has some issues, and the justification for its use is a single study on individual variations within humans only. This study cannot be extrapolated across species. Indeed, the olfactory bulb should be at least in part a function of the size of the animal (or of the nose more specifically), as a larger nose is filled with more olfactory mucosa, which thereby result in an increased OB size simply because that’s where the axons of the receptors all terminate. Second, the authors should have used a histological examination of OE surface areas, because simply comparing the sizes of the superior/posterior turbinates are not enough, as it is known the different species have different percentages of a given turbinate covered in OE.

Identifying correlates of the olfactory sense is indeed difficult and an ongoing endeavor in sensory ecology. The approach we take here is to assess commonly used approaches as our goal is to test the extent to which commonly-used anatomical metrics correlate with behavioural proxies of

olfaction. Both MOB size and nasal turbinate surface area are common proxies of olfaction used in extant taxa and extinct taxa, including rodents (Martinez et al. 2018, 2020), carnivorans (van Valkenburgh et al 2011, 2014, Green et al. 2012), and primates (Barton 1995, 2006; Heritage 2014; Smith 2014; Lundeen & Kirk 2019). Using the *relative* (brain, cranial or body-mass scales) MOB and nasal turbinate surface area is a common approach, and we agree with the reviewer that this helps control for the potentially confounding problem of having a larger nose due to having a larger body. On the other hand, many researchers (not just the study of intraspecific variation in humans) use absolute MOB and nasal turbinate surface area and some have argued that absolute MOB is the most appropriate proxy of olfactory sensitivity (see discussions in Smith & Bhatnagar 2004 and Heritage 2014). We cite several taxonomically broad studies taking one or both of these approaches in the introduction (lines 109-111), including two phylogenetically-controlled studies of primate MOBs (Barton 2006; DeCasien & Higham 2019), a review of comparative primate nasal anatomy (Smith et al. 2007), and a study of ethmoid areas across 32 mammals (Garrett & Steiper 2014). To this list, we have now also added Green et al. (2012) as an additional example, as they specifically investigated nasal turbinate (turbinal) surface area as an olfactory proxy in relation to diet across 20 carnivoran species. Because using relative (correcting for brain volume or skull size) *and* absolute MOB and nasal turbinate surface area measures are widespread in the literature, we chose to assess both here. We feel this approach will be informative to researchers studying anatomical correlates, allow us to contextualize our results with previous findings and hypotheses, and to identify any discrepancies in the results that might be informative and inspire future research.

We emphasize that we are not testing the legitimacy of these proxies. Rather, we are looking at two common metrics in the context of our results and noting the discordance between them as potentially interesting. We have now clarified this in Discussion (c)

lines 420-424: Rejection of specific anatomical features as behavioural proxies would be premature. However, our results highlight the importance of recognizing that senses are multifaceted and that different metrics may measure different dimensions of a given sense (e.g., threshold sensitivity vs. discrimination vs. breadth of stimuli [72]).

Additionally, we agree it would have been wonderful to include histology. However, lack of appropriate samples precludes this at this time. We have mentioned this as a promising future direction (line 427).

2. Regarding the correlates of taste (lines 189-190), is the anterior 0.5 cm of the tongue the same *relative* amount of the tongue across the 3 species? It would seem like this should be standardized, and I'm sorry if I missed that it was.

Unfortunately, we used previously published data for this analysis, and do not have data on lingual surface area across the three species. However, given that the tongue also scales with body size, we did include an analysis of fungiform papillae density relative body mass in our results (Methods: lines: 223-224; Results: lines: 291-295). The interspecific patterns were the same, with the addition of a significant difference between *Cebus* and *Ateles* when body mass is controlled for. We have modified the paragraph on data analyses to make our approach clearer:

Lines 221-224: To control for scaling effects of body size, we also calculated “size-corrected” measures of sensory anatomy. For nasal turbinate surface area, we correct for skull size. Tongue area was not available for studied species, so we control for body mass when calculating relative tongue papillae density.

In addition, to better emphasize the inclusion of the analyses of relative size, we have revised our Table S4 to include values from all analyses. We have also now added a new Supplementary Figure (new Figure S3) showing variation within genera for the two relative metric analyses for which we have multiple data points per genus (relative nasal turbinate surface area and relative fungiform papillae density) with an expanded caption summarizing why we performed these analyses, how the metrics were calculated, the results of statistical analyses, and how the results compare to those based on the raw metrics.

3. The findings in Question 2 are interesting, but I was hoping to see purely a (statistical) comparison between di- and tri-chromatic *Ateles* and *Cebus* only—i.e. without *Alouatta*, because it seems like the data from the howler monkeys could influence the results in a disjointed way, since they are only represented as tri-chromatic animals. Did the authors do this and I’ve missed it?

We did perform an analysis including only *Ateles* and *Cebus* for Question 2, for the reasons the reviewer mentions. In the previous version, this was mentioned in the results, but not the methods. We have made revisions to help clarify this (below) and provide more details in the Supplementary Information.

Methods Lines 266-268: “To address possible confounding influences of sex and routine vs. polymorphic trichromacy on these analyses, we repeated this analysis with only females of polymorphic species (*Ateles*, *Cebus*).”

Results lines 326-328: When we repeated this analysis with only females of polymorphic species (*Ateles*, *Cebus*), we identified the same pattern (Supplementary Results S3, Table S6, Fig. S3), but also identified significant main effects of fruit conspicuity on the use of manual touch and fruit acceptance. Among *Ateles* and *Cebus* females, cryptic fruits were touched more often and were accepted less often relative to conspicuous fruits.”

4. Unfortunately, Discussion point “c” is too heavy-handed based on the results of this paper. The paper has various strengths, but taxonomic breadth is not one of them. Thus, I don’t think the results from this study are enough to call into question, e.g., proxies used for olfaction. For example, larger MOB volume is *not* associated with active sniffing (Line 411), and I don’t even know what such a statement would mean. Further, larger turbinate SA has been associated with many aspects of olfaction, and the support for it being related to sensing faint odorants is very scant and not conclusive (again, human studies are not really appropriate in this context). Larger turbinate SAs could also be related to the breadth of odors that can be sampled, as that surface gives more space for a sufficient quantity of diverse OR receptors to be expressed. Whatever the case, we’re still on very tentative ground here, and this study doesn’t really change that.

We appreciate this feedback, which has inspired us to make substantive revisions to avoid over-interpreting our results. It was not our intention to call into question anatomical proxies because they do not mirror the behaviour metrics. Rather, we hoped to emphasize that these different anatomical and behavioral metrics could be measuring different dimensions of a given sense. We have rewritten large sections of the discussion to make this clearer, to highlight that olfactory perception is influenced by many variables and remains poorly understood. We briefly contextualize our findings and end with a call for future research. (Lines 337-360 and Lines 416-429)

Other Specific Comments

Lines 67-74: There's a substantial literature on fruits and olfaction in rodents, no?

Although there is research on the role of olfaction in scatter-hoarding in rodents, we are unaware of a substantial body of literature on olfaction and frugivory in wild rodents to the extent seen in bats. However, we have modified our statement in the Introduction to include this research, which we agree is important.

“A large body of work has explored the role of colour vision in finding and assessing foods in terrestrial vertebrate taxa [3–5] but the role of non-visual senses during foraging has received considerably less attention, with the notable exception of chiropterans [6], and a growing literature on primates and scatter-hoarding rodents [7-9] ”

Line 149: Was sniffing simply coded as binary (i.e. occurred or not), or was the duration of sniffing taken into account?

Sniffing was scored as binary, and we have revised the methods to clarify this (lines 149):

“Sniffing was coded as present (yes/no) if fruits were brought close to or in contact with the nose (Supplementary Video 1, Supplementary Video 2).”

Discussion: It would be nice for the authors to have a preliminary paragraph that summarizes the major points, and then they go into detail later. I don't really think that breaking the discussion down into Question 1 and Question 2 is necessary.

Space limitations motivated us to not include a summary paragraph, especially given requested revisions to other sections. We prefer to keep the structure of the discussion centered around our research questions to maintain the framework we establish in the introduction. However, we have removed the reference to the question number (“Question 1, 2”), as we agree it was unnecessary and distracting, and now use just the topic of the question as our heading.

Lines 336-338: What does this sentence actually mean? i.e. the MOB *always* processes odors, during “active sniffing” or not, so I'm really not sure what the authors are trying to say here.

We have removed this statement and rewritten this section. Please see our response to point #4 raised above.

Lines 341-342: I think the authors are misinterpreting this paper. Passive smelling of course activates brain regions that differ from active sniffing-induced smelling. But, both types of smelling always activate the MOB. So, I'm not sure what the authors are arguing for here, either.

We have removed this statement and rewritten this section.

Throughout: Please make sure to use italics for genus/species names throughout. Also, in my opinion it's better to just use the Scientific names rather than going back and forth between scientific names and common names, as much of the readership may not be familiar with primates at all.

We have revised our manuscript throughout to appropriately use italics, and we now primarily use scientific names, with only occasional use of common names (only at the start of the introduction and discussion in order to comprehensively introduce our study species in each section).

Figure 2: I find these plots very confusing. Are the other, more intuitive ways of plotting these data?

We considered several ways to represent the data, and settled on these plots as the most effective. However, we have revised the caption of Figure 2 to facilitate improved comprehension.

Journal Name: Proceedings of the Royal Society B

Journal Code: RSPB

Print ISSN: 0962-8452

Online ISSN: 1471-2954

Journal Admin Email: proceedingsb@royalsociety.org

MS Reference Number: RSPB-2022-0048

Article Status: REJECTED

MS Dryad ID: RSPB-2022-0048

MS Title: Anatomy and dietary specialization influence sensory behaviour among sympatric frugivorous primates

MS Authors: Melin, Amanda; Veilleux, Carrie; Janiak, Mareike; Hiramatsu, Chihiro; Sánchez-Solano, Karem; Lundeen, Ingrid ; Webb, Shasta; Williamson, Rachel; Mah, Megan; Murillo-Chacon, Evin; Schaffner, Colleen; Hernández-Salazar, Laura; Aureli, Filippo; Kawamura, Shoji

Contact Author: Amanda Melin

Contact Author Email: amanda.melin@ucalgary.ca

Contact Author Address 1: 2500 University Drive NW

Contact Author Address 2:

Contact Author Address 3: Calgary

Contact Author City: Calgary

Contact Author State: Alberta

Contact Author Country: Canada

Contact Author ZIP/Postal Code: T2N 1N4

Keywords: Sensory ecology, plant-animal interactions, colour vision, olfaction, frugivory, platyrrhine

Abstract: Senses form the interface between animals and environments, and their form and function provide a window into the ecology of past and present species. However, research on the senses used during foraging by wild terrestrial vertebrates (smell, taste, vision, and touch) is sparse. Here, we combine 26,094 foraging sequences recorded from three wild, sympatric primate species (*Cebus imitator*, *Ateles geoffroyi*, *Alouatta palliata*) with measurements of sensory anatomy. We hypothesize that dietary and sensory specialization shape foraging behaviours used during fruit assessment. We find the most frugivorous species (*Ateles geoffroyi*) sniffs fruits the most often, the species with the highest measure of manual dexterity (*Cebus imitator*) uses manual touch most often, and that main olfactory bulb volume is a better predictor of sniffing behaviour than nasal turbinate surface area. We also identify an interaction between colour vision phenotype and use of other senses. Controlling for species differences, dichromats sniff and bite fruits more often than trichromats do, and trichromats use manual touch to evaluate cryptic fruits more often than dichromats do. Our findings shed light on how dietary specialization and sensory variation shape foraging behaviors, and on methods for investigating the relationships between behaviour and anatomy.

EndDryadContent

Appendix D

Professor Loeske Kruuk

Dear Dr Melin

I am pleased to inform you that your manuscript RSPB-2022-0847 entitled "Anatomy and dietary specialization influence sensory behaviour among sympatric frugivorous primates" has been accepted for publication in Proceedings B. My apologies for the delay in returning a decision to you. The reviewer and Associate Editor's comments (not including confidential comments to the Editor) are included at the end of this email for your reference, as well as a couple of minor ones from me.

Dear Professor Kruuk,

We are very pleased to receive this welcome decision on our paper, and we very much appreciate your additional constructive feedback. We reply in detail to the comments from yourself and the other editors and reviewers, below.

The paper is much improved, but there is still some concern about its limitations given the small number of species used. The Associate Editor who reviewed this version has also made some very important points with regard to the results being clear for a non-specialist (clarifying points that I was certainly finding puzzling). We would therefore like to invite you to revise your manuscript to address these issues. Toning down any general conclusions regarding different types of species is essential. In many cases, this can be done very simply by making it clear that you are just referring to the species considered here (e.g. L362, change 'We predicted that species more reliant on ripe fruit would...' to 'We predicted that the species more reliant on ripe fruit (species X and species Y) would...'). Please check through carefully for any other such instances, and please address the AE's comment below about acknowledging the limitations of only using three closely-related species.

We are grateful for the detailed suggestions. We have revised our manuscript accordingly to clarify the study system, sample size, and specific results with respect to predictions (changes/ additions in blue text). A list of the specific changes is found below, and also found in the appropriate sections in response to feedback from the reviewers and Associate Editor.

Abstract

Lines 52-54: Here, we combine 26,094 fruit foraging sequences recorded from three wild, sympatric primates (*Cebus imitator*, *Ateles geoffroyi*, *Alouatta palliata*) with data on within- and between-species variation in colour vision, olfaction, taste, and hand anatomy.

Introduction

Line 115: Here, we examine three species of wild sympatric primates to address two questions:

Lines 125-129: Does variation in colour vision phenotype (dichromatic vs. trichromatic vision) influence the use of other sensory behaviours and food selection accuracy? Because two of the three primate species exhibit intraspecific variation via polymorphic trichromacy, we can investigate the relationships between colour vision and non-visual senses, while controlling for species-level variation.

Discussion

Lines 339-342: Our results offer mixed support for our hypothesis regarding dietary specialization. As predicted, the highly frugivorous *Ateles* sniffed fruits the most frequently; however, *Cebus* monkeys, the next most frugivorous species, sniffed fruits less frequently than the least frugivorous howler monkeys (*Alouatta*).

Lines 352-354: Our data, although limited to three species, suggests that nasal turbinate volume may not be useful for predicting active sniffing in a fruit foraging context, and might be a better metric of other aspects of olfaction, an idea that invites future study.

Lines 358-360: While we examined a narrow ecological context (e.g., active sniffing by three species in the context of fruit foraging), our data suggest complicated relationships between diet, olfactory structures, and behaviour.

Lines 364-366: We predicted that species more reliant on ripe fruit (i.e. *Ateles*, and to a lesser extent *Cebus*) would use manual touch more often than less frugivorous species, as fruit softening is a reliable cue of ripening for many plant species [16].

Lines 379-382: The evolution of this derived manual touch behaviour suggests that palpating fruit is informative to spider monkeys and adaptations favoring discriminative touch may be under selection in frugivorous primates more broadly.

Lines 404-406: We also demonstrate for the first time that dichromatic platyrrhine primates bite and reject fruits more frequently than trichromats.

Lines 413-416: Regardless, we find that variation in one sensory system (colour vision) influences the use of multiple other senses when investigating the same fruit types among three sympatric platyrrhine primates, highlighting the need for future studies exploring multimodal sensory integration across additional diverse species.

Lines 423-424: For example, of the primates we examined, we found species that used sniffing behaviours more often had relatively large MOBs, but not relatively large nasal turbinates.

Because the schedule for publication is very tight, it is a condition of publication that you submit the revised version of your manuscript within 7 days. If you do not think you will be able to meet this date please let us know.

Thank you again. We have worked hard to meet this deadline and hope our paper is now found suitable for publication.

Associate Editor

Board Member: 1

Comments to Author:

I am seeing this paper for the first time as a reviewer. In general, I like it quite a lot; its a neat system and I am pleased to see the authors comparing three sympatric species. I have just a couple of big picture comments that are needed to make this manuscript ready for publication.

First, while this is a fantastic system, at times the authors are overstretching their conclusions; it is only three species and more than that, it's three very closely related platyrrhines. How do we know that these results will generalize to the rest of the primates, much less other mammals or animals? A bit more circumspection is needed when interpreting these results.

We are grateful for these comments. We have revised our manuscript to be clearer about our study system and its limitations. We made the following revisions (changes/ additions in blue).

Introduction

Line 115: Here, we examine three species of wild sympatric primates to address two questions:

Discussion

Lines 339-342: As predicted, the highly frugivorous *Ateles* sniffed fruits the most frequently; however, *Cebus* monkeys, the next most frugivorous species, sniffed fruits less frequently than the least frugivorous howler monkeys (*Alouatta*).

Lines 352-354: Our data, although limited to three species, suggests that nasal turbinate volume may not be useful for predicting active sniffing in a fruit foraging context, and might be a better metric of other aspects of olfaction, an idea that invites future study.

Lines 358-360: While we examined a narrow ecological context (e.g., active sniffing by three species in the context of fruit foraging), our data suggest complicated relationships between diet, olfactory structures, and behaviour.

Lines 364-366: We predicted that species more reliant on ripe fruit (*i.e.* Ateles, and to a lesser extent *Cebus*) would use manual touch more often than less frugivorous species, as fruit softening is a reliable cue of ripening for many plant species [16].

Lines 379-382: The evolution of this derived manual touch behaviour suggests that palpating fruit is informative to spider monkeys and adaptations favoring discriminative touch may be under selection in frugivorous primates more broadly.

Lines 404-406: We also demonstrate for the first time that dichromatic platyrrhine primates bite and reject fruits more frequently than trichromats.

Lines 413-416: Regardless, we find that variation in one sensory system (colour vision) influences the use of multiple other senses when investigating the same fruit types among three sympatric platyrrhine primates, highlighting the need for future studies exploring multimodal sensory integration across additional diverse species.

Line 423-424: For example, of the primates we examined, we found species that used sniffing behaviours more often had relatively large MOBs, but not relatively large nasal turbinates.

Second, quite a lot of the paper is written as if for a specialist audience, which means that some of the most interesting components are not going to be evident to the average reader. For instance, nowhere in the abstract do you mention the unusual color vision system of the *Cebus* and *Ateles* monkeys, which makes them a particularly good pair of species to study as you can compare di- and tri-chromats within-species. This oversight is curious and seems to assume that the reader is sufficiently familiar with this system that it would be obvious. Please read over your paper carefully to ensure that it is written so that you neither overstep the limits of your data nor make unwarranted assumptions about what your reader already knows; after this is done I think this will be a nice addition to the literature.

We appreciate this feedback. We have now revised our manuscript to help avoid the assumption of prior knowledge of the study system, we have added mention of within and between species variation in colour vision to the abstract, and made other minor edits to allow us to space within the 200 word limit. We regret not being able to explain the polymorphism in more depth due to space constraints, but we believe this now better flags the interesting variation in the abstract. We add additional explanations in the main text, as described below (changes and additions are in blue)

Abstract

Lines 52-54: Here, we combine 26,094 **fruit** foraging sequences recorded from three wild, sympatric primates (*Cebus imitator*, *Ateles geoffroyi*, *Alouatta palliata*) with **data on within- and between-species variation in colour vision, olfaction, taste, and hand anatomy**.

Introduction

Lines 125-129: **Does variation in colour vision phenotype (dichromatic vs. trichromatic vision) influence the use of other sensory behaviours and food selection accuracy?** Because two of the three primate species exhibit intraspecific variation via polymorphic trichromacy, we can investigate the relationships between colour vision and non-visual senses, while controlling for species-level variation.

Editor (LK) minor comments

Where you cite a statistic, be sure to include the relevant d.f., e.g. for the chi-squared values at L274.

We are very grateful to the editor for catching this error. We have now revised this, and have checked all of the other statistics in the main text and supplemental and confirmed that they include degrees of freedom as a subscript to X^2 .

Typo at end of L289.

Thank you. There was a conversion error for some of the chi-squared symbols in the Results section. We have corrected these.

L338 - check syntax. (Consider rewriting this sentence, as it is not easy to follow.)

We have rewritten this section as follows (now lines 339-341):

As predicted, the highly frugivorous *Ateles* sniffed fruits the most frequently; **however,** *Cebus* monkeys, the next most frugivorous species, sniffed fruits less frequently than **the least frugivorous** howler monkeys (*Alouatta*).

We also revised this sentence to improve clarity:

Lines 87-90: Although sparse, current literature suggests that primates differ in sensory use [10,24–26]. For example, **when presented with the same novel stimuli**, spider monkeys (*Ateles geoffroyi*) were more likely to **sniff stimuli**, while squirrel monkeys (*Saimiri sciureus*) were more likely to touch **them** [25].

Reviewer(s)' Comments to Author:

Referee: 3

Comments to the Author(s).

I appreciate the revised version of the manuscript that the authors submitted, and I find it to be very strong still in a number of areas. I think it approaches the field of comparative sensory biology from a novel and robust perspective, and for this reason it makes a valuable contribution.

We are grateful for the supportive feedback.

My final remaining comment still concerns how to interpret these results in the broader context of the field. For examples, lines 350-351 and 377-379 are still a bit heavy-handed, in my opinion. But, overall I think this is a very minor critique and I am happy with how the authors have contextualized their findings and offered suggestions going forward.

We have modified our text to be less heavy-handed and to highlight the limitations of our study, detailed below.

Lines 350-354 (previously 350-351): “Interestingly, absolute and relative nasal turbinate surface areas were smallest in Ateles (an avid fruit sniffer), and largest for Alouatta (intermediate for sniffing behaviours). Our data, although limited to three species, suggests that nasal turbinate volume may not be useful for predicting active sniffing in a fruit foraging context, and might be a better metric of other aspects of olfaction, an idea that invites future study.”

Lines 379-382 (previously 377-379):

- *Original wording*: “The evolution of this derived manual touch behaviour supports the hypothesis that palpating fruit is informative for frugivores and likely under selection.”
- *New wording*: “The evolution of this derived manual touch behaviour suggests that palpating fruit is informative to spider monkeys and adaptations favoring discriminative touch may be under selection in frugivorous primates more broadly.”

Lines 421-422: “Our results suggest that we need to be thoughtful~~think critically~~ in interpreting ~~about how~~ commonly-used proxies of sensory reliance ~~are interpreted.~~”

Appendix E

Response to editor

Comments from Editor Loeske Kruuk

Thanks for submitting a revised version of this manuscript, and for the changes you have made to address the previous comments.

I was hoping to be able to accept this version without any further changes. However, I'm afraid that the Abstract is still not addressing the problem of explaining the system clearly enough to readers who do not already know the details. In particular, it is not clear in the Abstract that capuchins and spider monkeys are both di- and tri-chromats. I appreciate that you say at the start of the Abstract that you use within-species variation on a range of traits, and that you mention 'controlling for species', but the abstract needs to say explicitly that there is a polymorphism in vision type within species in your study species. Please refer back to the explicit comments on the previous version from the AE about not having mentioned the polymorphism in the abstract: 'This oversight is curious and seems to assume that the reader is sufficiently familiar with this system that it would be obvious'.

Please therefore revise your manuscript to address this important issue with the abstract (I suggest giving it to non-experts in the field to test if the polymorphism aspect is clear to them.) Because the schedule for publication is very tight, it is a condition of publication that you submit the revised version of your manuscript within 7 days. If you do not think you will be able to meet this date please let us know.

Dear Editor Kruuk,

Please accept our apologies that our previous revision to the abstract was not sufficient. We have now rewritten our abstract to address this request, while altering to adhere to space constraints. Thank you for this opportunity. We hope our revised manuscript is now found to be suitable for publication.

Amanda Melin, on behalf of all coauthors.